# Partner choice and cooperation in social dilemmas can increase resource inequality

Mirre Stallen [1,2,5], Luuk L. Snijder [1,5] ✉, Jörg Gross [3], Leon P. Hilbert [4] & Carsten K. W. De Dreu [1]

Cooperation is more likely when individuals can choose their interaction partner. However, partner choice may be detrimental in unequal societies, in which individuals differ in available resources and productivity, and thus in their attractiveness as interaction partners. Here we experimentally examine this conjecture in a repeated public goods game. Individuals ($n = 336$), participating in groups of eight participants, are assigned a high or low endowment and a high or low productivity factor (the value that their cooperation generates), creating four unique participant types. On each round, individuals are either assigned a partner (assigned partner condition) or paired based on their self-indicated preference for a partner type (partner choice condition). Results show that under partner choice, individuals who were assigned a high endowment and high productivity almost exclusively interact with each other, forcing other individuals into less valuable pairs. Consequently, pre-existing resource differences between individuals increase. These findings show how partner choice in social dilemmas can amplify resource inequality.

In social dilemmas, overall welfare is maximized if everyone cooperates, while individuals are also tempted to defect and free-ride on others' cooperation. These dilemmas occur at all levels of society, from small groups to nations[1]. Cooperating in social dilemmas is crucial for the functioning of groups and society at large[2,3]. Conversely, defection in social dilemmas has been linked to group dissolution, polarization, and conflict[1,4].

Cooperation is more likely when individuals can choose their interaction partner[5–11]. Under partner choice, individuals willing to cooperate can find others also willing to cooperate, thereby avoiding the risk of being exploited by defectors. Indeed, computational models and experiments revealed how partner choice can lead to a segregation of the population into cooperating individuals on the one hand, and defectors on the other[5–12]. When this happens, defectors have an interest to switch to cooperation, since mutual cooperation is more beneficial than mutual defection. Through this mechanism, partner choice can provide a solution to the social dilemma of cooperation.

Past experimental work on partner choice in social dilemmas typically assumed that individuals have the same ability to reciprocate cooperation (but see[13,14]). However, apart from cooperation, people also use other cues, such as endowment and productivity, to select partners[5–8,10]. Indeed, individuals differ in the ex-ante resources they have available for cooperation, and in their capacity to produce joint wealth through cooperation, for example because they lack training or task-relevant experience[13,15,16]. As a result, and regardless of their willingness, some individuals can contribute more to public goods than others and are better positioned to reciprocate cooperation.

Recent work started to address how inequality affects cooperation in social networks[14], but how partner choice affects cooperation in social dilemmas when people have unequal access to resources or differ in their productivity remains an open question. Partner choice may lead highly endowed and highly productive individuals to seek out similar partners to benefit from cooperation, while less endowed and less productive individuals are excluded (i.e., segregation). Whereas in

[1]Social, Economic and Organisational Psychology, Leiden University, Leiden, The Netherlands. [2]Poverty Interventions, Center for Applied Research on Social Sciences and Law, Amsterdam University of Applied Sciences, Amsterdam, The Netherlands. [3]Institute of Psychology, University of Zurich, Zurich, Switzerland. [4]Institute of Psychology, University of Amsterdam, Amsterdam, The Netherlands. [5]These authors contributed equally: Mirre Stallen, Luuk L. Snijder. ✉e-mail: l.snijder@fsw.leidenuniv.nl

equal societies defectors can try to join communities of cooperators by switching from defection to cooperation, in unequal societies the ones who have less resources and are less productive cannot. Even if willing to cooperate, these individuals lack the capital or training to make themselves attractive partners for cooperative exchange. Whereas in past work partner choice emerged as a promising mechanism for fostering cooperation, partner choice may further exacerbate resource inequality when people have unequal access to resources or differ in their productivity.

We examined this possibility experimentally in groups of eight individuals. Participants ($n = 336$) interacted in a two-person public goods game in which cooperation maximized joint outcomes and free-riding maximized personal gains. Participants differed in their endowment and productivity factor, operationalized as the resources they could contribute to the public good and the value that their cooperation could generate, respectively (Fig. 1a; see also[13]). Within each eight-person group, two participants were randomly assigned to have a high endowment and high productivity factor (henceforth HH types), two were given a high endowment and low productivity factor (HL types), two had a low endowment and high productivity factor (LH types), and two were assigned a low endowment and low productivity factor (LL types; Fig. 1a). Individuals interacted in pairs for 24 rounds. In each round, participants within one pair simultaneously decided how much of their resources to contribute to their public good. Each unit was multiplied by participants' individual productivity factor and then distributed equally across the pair (Fig. 1b). At the end of each

round, participants learned how many units their partner allocated to the public good and about their earnings.

To investigate how partner choice influences cooperation, segregation, and resource inequality we manipulated (between groups) how pairs were formed (Fig. 1c). In the assigned partner condition, participants were assigned a partner on each new round. The assignment was pseudo-random so that participants interacted six times with each possible partner type. In the partner choice condition, participants were asked to rank partner types from most to least preferred at the start of each round. Participants were paired with their first choice if possible. If participants could not be paired with their first choice because no participants of their preferred partner type preferred to be paired with their type, they were paired with their second partner type choice, and so on (see Methods for more information).

In our experimentally induced 'unequal society', three possible patterns of partner selection and cooperation could emerge. One possibility rests on the idea that people are (highly) inequality averse, meaning that they prefer equal to unequal outcomes (especially when individual characteristics are based on luck, like the type assignment in our experiment)[17–19]. In theory, being free to choose a partner allows individuals to reduce resource inequality based on endowment and productivity by using the public good as a redistribution device between those who are disadvantaged (individuals assigned to a low endowment and low productivity factor) and those who are advantaged (individuals assigned to a high endowment and high productivity factor). If motivations to reduce outcome inequality are

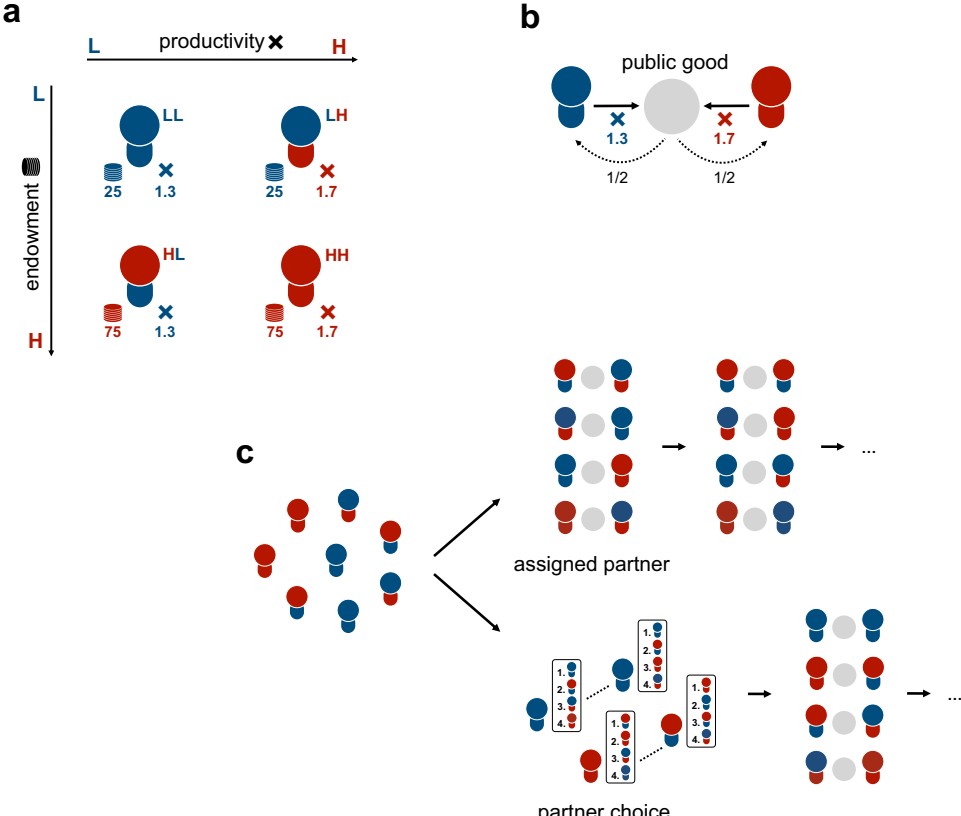

**Fig. 1 | Partner choice under inequality. a** Participants differed on two characteristics: endowment (H = 75 Units, L = 25 Units) and productivity (H = 1.7, L = 1.3), creating four different participant types: high endowment and high productivity (HH), low endowment and high productivity (LH), high endowment and low productivity (HL), and low endowment and low productivity (LL). **b** In each round, two participants were paired and decided how many units of their endowment to contribute to their shared public good. Each contributed unit was multiplied by the productivity factor of the participant and the resulting investment was divided

equally between the pair. **c** Between treatments, we manipulated how pairs were formed. In the assigned partner condition, participants were pseudo-randomly assigned to different types across rounds, such that everyone interacted with each partner type equally often. In the partner choice condition, participants, on each round, first indicated their partner preference by ranking all types from 1 (most preferred) to 4 (least preferred). Based on their ranking, participants were paired with a partner to play the public goods game with, for that round.

strong enough, we should see that especially those individuals who are advantaged in terms of their assigned endowment and productivity factor want to be paired and cooperate with partners who are disadvantaged in our random assignment of types. In this scenario, we should not observe a segregation of populations based on endowment and productivity levels.

An alternative to this possible pattern stems from the idea that individuals have homophily preferences – they prefer similar to dissimilar partners, and cooperate more with similar others[20–23]. Such homophily preferences may be grounded in empathy avoidance[24–26] or in avoiding the (psychological) costs of rejection by individuals assigned to a high endowment and/or high productivity factor[6]. Homophily preferences have been proposed in theoretical models[27] and are supported by data on demographic compositions of neighborhoods[28] and ethnic demarcation[29]. In our experiment, partner selection based on homophily would lead to disproportionate pairings of similar (e.g., HL-HL and LH-LH) rather than dissimilar types. Homophily can thus endogenously create segregation with some pairings (such as HH-HH) being better positioned to generate welfare through cooperation than other pairings (such as LL-LL). Partner choice based on homophily preferences would thus exaggerate resource inequality.

The third possible pattern of results is based on the idea that people, in general, prefer to be partnered with others who are advantaged (i.e., partners assigned a high endowment and high productivity) instead of with those who are disadvantaged (partners assigned a low endowment and low productivity), as cooperating with the former more likely generates value to themselves[7,10,14]. However, in unequal societies, and in our experiment, not all individuals can be paired with their most preferred partner. If individuals assigned to a high endowment and/or high productivity factor are indeed most preferred by everyone, partner choice in combination with a population-wide preference for those types would lead to a segregation of pairs consisting of individuals assigned to a high endowment and high productivity factor versus the rest. Because these highly endowed and highly productive pairs are particularly well suited to generate welfare through cooperation, such segregation would also amplify resource inequality over time.

In line with this third possibility, we show that partner choice exacerbates pre-existing resource differences between individuals assigned to a high endowment and high productivity factor, and individuals assigned to a low endowment and low productivity factor. Individuals who are advantaged in terms of their endowment and productivity factor almost exclusively prefer to interact with each other. As a result, individuals who are endowed with less and are unable to generate many resources because of their lower productivity factor become forced to work together and also cooperate less. Over time, this process increases pre-existing resource differences between individuals.

## Results

### Segregation under partner choice

In the assigned partner condition, pairs were uniformly distributed by design, meaning that participants interacted an equal number of times with each type (Fig. 2a). This pattern did not evolve in the partner choice condition. Participants were paired with their first choice in 45% of the rounds and we observed more same-type pairs in the partner choice condition than in the assigned partner condition (diagonal in Fig. 2a, b; multilevel logistic model [MLLM], $z = -7.56$, $b_{condition} = -1.68$, $p < 0.001$, 95% CI [−2.18, −1.19], Supplementary Table 1). Also, same-type pairs were more stable, in that they interacted for more consecutive rounds on average (Fig. 2c, multilevel model [MLM], $t(292) = 12.14$, $b_{similar} = 2.77$, $p < 0.001$, 95% CI [2.32, 3.23], Supplementary Table 2). Thus, partner choice led to a segregation of the population into 'homogenous neighbourhoods' in which similar types almost exclusively interacted with each other.

The segregation observed in the partner choice condition coincides with the possibility that individuals prefer to be paired with others of their own type (i.e., they have homophily preferences) or that there is a population-wide preference for HH types. This latter mechanism was further supported when we examined which types participants preferred to be paired with (in line with our pre-registration). Participants who were assigned a high-endowment high-productivity type were the most preferred partner types (Fig. 3a; i.e., most popular; across all types, 65.1% of all first choices was an HH type; MLLM, $z = 9.76$, $b_{type} = 5.56$, $p < 0.001$, 95% CI [4.25, 6.86], Supplementary Table 3). In contrast, participants who were assigned a low-endowment low-productivity type were rejected most often, meaning that they could not be paired with the partner type of their first choice in most (75%) of the rounds (Fig. 3a; MLLM, $z = 6.34$, $b_{type} = 1.85$, $p < 0.001$, 95% CI [1.18, 2.53], Supplementary Table 4). Participants even tried to avoid partners who were assigned an LL type (i.e., participants preferred to be paired with a different type than an LL type, after being paired with an LL type in the previous round; MLLM, $z = 5.67$, $b_{previouspartner} = 0.72$, $p < 0.001$, 95% CI [0.43, 1.01], Supplementary Table 5).

As a result of these partner preferences, the most prevalent pairing in the partner choice condition consisted of two HH types being paired together; in 73% of the rounds a participant who was assigned an HH type was paired with another HH type (Fig. 2b). Although most participants did not want to be paired with an LL type, the second most prevalent pairing was between two LL types; in 66.9% of the rounds a participant who was assigned an LL type was paired with another LL type (Fig. 2b; note that this pattern was more pronounced in the 16 groups recruited via Leiden University, than in the five groups recruited via Prolific, see Methods and Supplementary Note 1). Hence, HH and LL types were more likely to be paired with their own type than HL and LH types were (MLLM, $z = 8.74$, $b_{type} = 2.15$, $p < 0.001$, 95% CI [1.60, 2.70], Supplementary Table 1) such that segregation occurred by clustering of the extremes. However, because participants who were assigned an LL type did not prefer to be paired with their own type (only in 17.3% of all rounds), they were not paired by choice. Instead, these LL types were paired disproportionately often because no other type in the population wanted to be paired with them. Partner choice thus not only promoted segregation, but also led to unequal success rates in interacting with one's partner of choice, further disadvantaging the disadvantaged.

Partner rankings changed over time, so that apparent homophily preferences became more prevalent in later rounds (Fig. 3b). In the first round, most participants, regardless of their assigned type, preferred to be paired with an HH type. However, the preference for HH types decreased over time for participants who were assigned a different type (MLLM, $z = -9.24$, $b_{round} = -0.07$, $p < 0.001$, 95% CI [−0.08, −0.05], Supplementary Table 6), possibly because HH types were out of reach for the other types, or because non-HH types wanted to avoid rejection by HH types. On the other hand, we found no credible evidence suggesting that HH types switched partner preferences over time (MLLM, $z = 0.15$, $b_{round} = 0.002$, $p = 0.879$, 95% CI [−0.03, 0.04], Supplementary Table 7). Consequently, participants who were assigned an HL, LH, and LL type increasingly preferred partners of their own type (i.e., homophily; HL: MLLM, $z = 2.57$, $b_{HL \times round} = 0.06$, $p = .010$, 95% CI [0.01, 0.10]; LH: MLLM, $z = 1.98$, $b_{LH \times round} = 0.04$, $p = .048$, 95% CI [0.00, 0.08]; LL: MLLM, $z = 3.19$, $b_{LL \times round} = 0.07$, $p = .001$, 95% CI [0.02, 0.12], Supplementary Table 7). This suggests that partner choice can create homophily preferences over time, with rejection avoidance being a possible driver of this effect. Importantly, homophily emerged as a consequence rather than cause of population segregation.

### Partner choice, cooperation, and the distribution of resources

Although partner choice segregated the population by type, it also produced higher overall cooperation compared to the assigned partner condition (Fig. 4a). Specifically, relative cooperation (the average

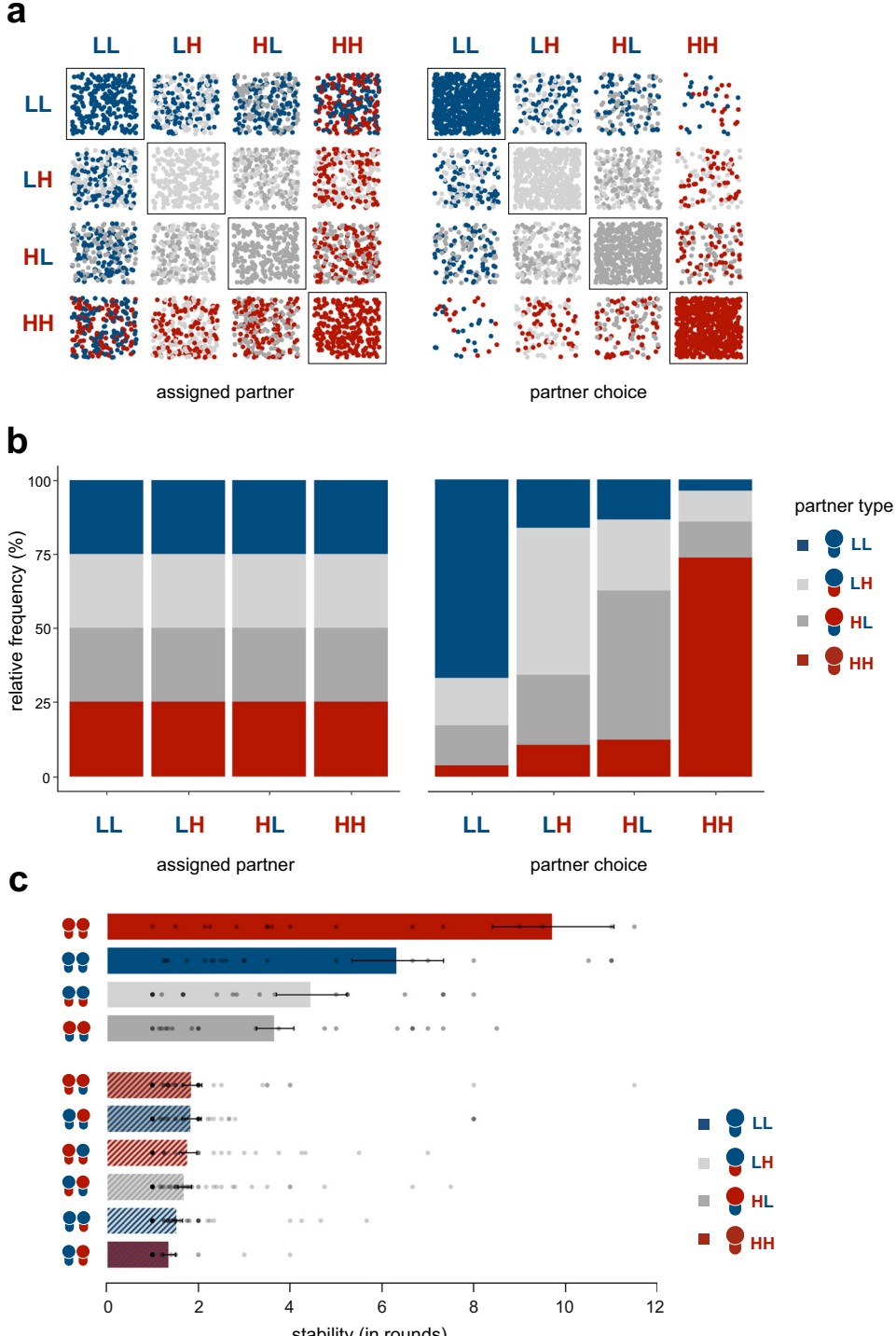

**Fig. 2 | Population segregation.** Red = high endowment and high productivity (HH), light grey = low endowment and high productivity (LH), dark grey = high endowment and low productivity (HL), blue = low endowment and low productivity (LL). **a** Spatial grid displaying the frequency of observed types in each possible pairing configuration in the assigned partner (left) and partner choice condition (right). Each dot represents one observation per type and pairing. For example, the top right block corresponds to a pairing configuration of an LL type interacting with an HH type: The blue dots represent the number of LL types, and the red dots represent the number of HH types that were in this pairing configuration in the assigned partner condition (left) or in the partner choice condition (right). All dots together reflect the number of participants that were part of an HH-LL pair. The

frequency of dots along the diagonal shows that partner choice led to a segregation of the population into pairs of similar types (higher frequency of dots on the diagonal in the partner choice condition than in the assigned partner condition). **b** Stacked bar graph illustrating the relative frequency of observed pairs in the assigned partner (left) and partner choice condition (right). Each bar represents the relative frequency with which a participant type was paired with their own or another type. **c** Average length of consecutive interactions between different pairs (as a measure of pair stability) in the partner choice condition ($n = 21$ groups). Whereas HH-HH pairs interacted 9.7 consecutive rounds on average, HH-LL pairs were least stable and only interacted 1.4 consecutive rounds on average before breaking up. Error bars indicate the standard error of the mean.

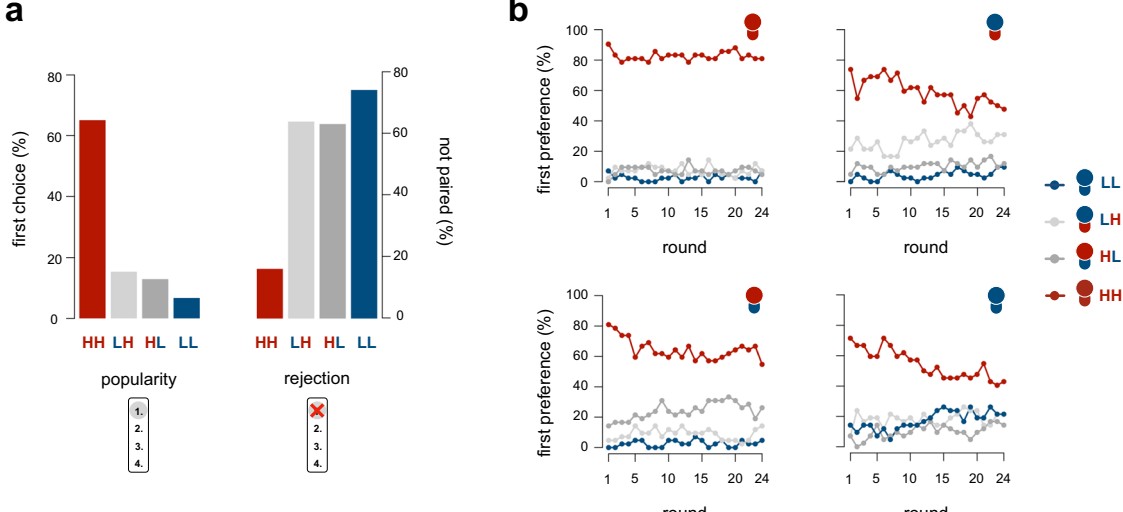

**Fig. 3 | Partner preferences in the partner choice condition.** Red = high endowment and high productivity (HH), light grey = low endowment and high productivity (LH), dark grey = high endowment and low productivity (HL), blue = low endowment and low productivity (LL). **a** The most popular partners (i.e., ranked as first choice) were types assigned a high endowment and a high productivity (HH) – left panel. Participants assigned a low endowment and a low productivity (LL) had the highest likelihood to not get paired with their preferred partner type (measure of rejection – right panel). **b** Time trend showing the development of participants' partner type preferences (first choices), depicted separately for each participant type.

contributions to the public good as a percentage of participant's individual endowment) decreased over time in the assigned partner condition (MLM, $t(7726) = -4.38$, $b_{condition × round} = -0.33$, $p < 0.001$, 95% CI [−0.48, −0.18], Supplementary Table 8), but we found no credible evidence suggesting that cooperation changed in the partner choice condition (MLM, $t(7726) = -1.54$, $b_{round} = -0.08$, $p = 0.125$, 95% CI [−0.19, 0.02]; Supplementary Table 8). These results, however, should be interpreted with some caution, because exploratory analyses showed that in the five groups recruited via Prolific there was an increase in cooperation over time in the partner choice condition, whereas cooperation decreased over time in the 16 groups recruited through Leiden University (see Supplementary Note 1). Nonetheless, findings resonate with previous work on partner choice showing that relative cooperation remained more stable under partner choice, possibly because participants could avoid uncooperative partner types. Indeed, results show that participants changed their partner preference if, on the previous round, their partner cooperated relatively less than they did (MLLM, $z = 3.67$, $b_{contribution} = 0.37$, $p < 0.001$, 95% CI [0.17, 0.57], Supplementary Table 9). This could also explain why the preference of non-HH types for HH types decreased over time (Supplementary Table 10). Participants also cooperated relatively more when their pairing was stable, with stability indicating the average length of consecutive interactions between paired types (MLM, $t(389) = 6.03$, $b_{stability} = 2.46$, $p < 0.001$, 95% CI [1.67, 3.25], Supplementary Table 11; controlling for type).

While, overall, partner choice countered the breakdown of cooperation, cooperation levels depended on partner type (Fig. 4a, see also Supplementary Fig. 1 and Supplementary Table 12). Participants cooperated relatively more when their partner was assigned an HH type (M = 70.75%, SE = 5.61) and relatively less when their partner was assigned an LL type (M = 50.83%, SE = 5.55). These differences were driven by whether participants were paired with their preferred partner type or not. Participants who were paired with the type of their first choice cooperated relatively more with their partner than those who were not paired with their first choice (MLM, $z = 13.28$, $b_{ranking} = 14.11$, $p < 0.001$, 95% CI [11.70, 16.52], Supplementary Table 13; controlling for own type, partner type, and round).

As a result of participants' partner preferences and their (relative lack of) cooperation with specific partner types, the a priori resource

gap that existed between types increased in the partner choice condition (Fig. 4b; in line with our pre-registration). Participants who were advantaged, i.e. who were assigned an HH type, accumulated more resources (i.e., total number of units at the end of the game) relative to participants who were assigned another type in the partner choice condition (MLM, $z = 49.09$, $b_{type} = 1345.93$, $p < 0.001$, 95% CI [1276.17, 1415.69], Supplementary Table 14). They also earned more than the participants who were assigned an HH type in the assigned partner condition (MLM, $z = 7.70$, $b_{condition} = 326.95$, $p < 0.001$, 95% CI [218.93, 434.97], Supplementary Table 14). Conversely, participants who were disadvantaged, i.e. who were assigned an LL type, accumulated less resources relative to participants who were assigned another type in the partner choice condition (MLM, $z = -41.05$, $b_{type} = -1125.44$, $p < 0.001$, 95% CI [−1195.19, −1055.68], Supplementary Table 14), and relative to participants who were assigned an LL type in the assigned partner condition (MLM, $z = -3.34$, $b_{condition} = -141.71$, $p = 0.005$, 95% CI [−249.73, −33.69], Supplementary Table 14).

We found no credible evidence suggesting that there were differences in population-level earnings between partner choice and assigned partners (MLM, $z = -1.17$, $b_{condition} = -36.04$, $p = 1.00$, 95% CI [−114.74, 42.67], Supplementary Table 14). At the same time, partner choice resulted in a much stronger skew in how accumulated earnings were distributed within the population (Fig. 4b). After the last round of the assigned partner condition, generated earnings were distributed slightly more equally across types (Gini index = 0.24) compared to types' starting positions (Gini index = 0.25), with participants assigned an HH type receiving 33.6% (starting position 38%) of all public goods earnings and participants assigned an LL type receiving 20.2% (starting position 12%). This skew was steeper in the partner choice condition (Gini index = 0.44), in which assigned HH types received 51.8% of all generated earnings and assigned LL types received only 9.0%. This shows that, being able to select with whom to cooperate can further increase pre-existing resource disparities when people differ in their endowment and productivity levels.

## Discussion

Past work revealed partner choice as a promising mechanism to promote cooperation in equal societies. Here we show that in artificial unequal societies, individual discretion to select whom to interact with

## a  cooperation

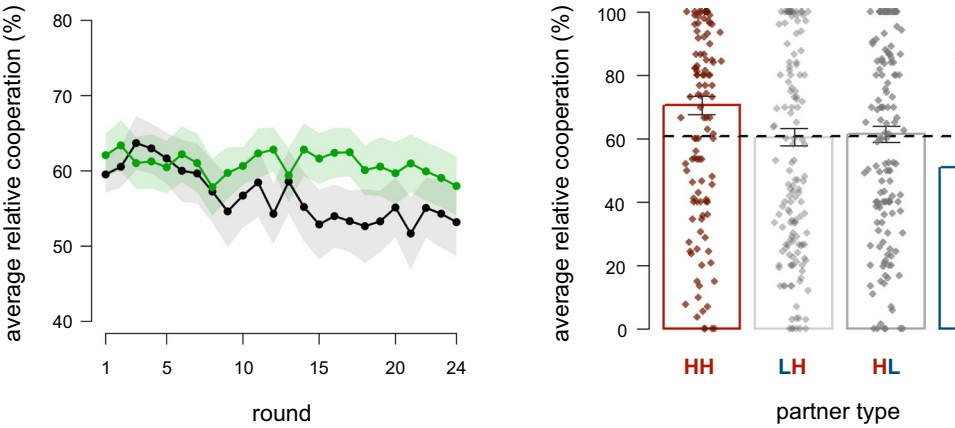

## b  resource distribution

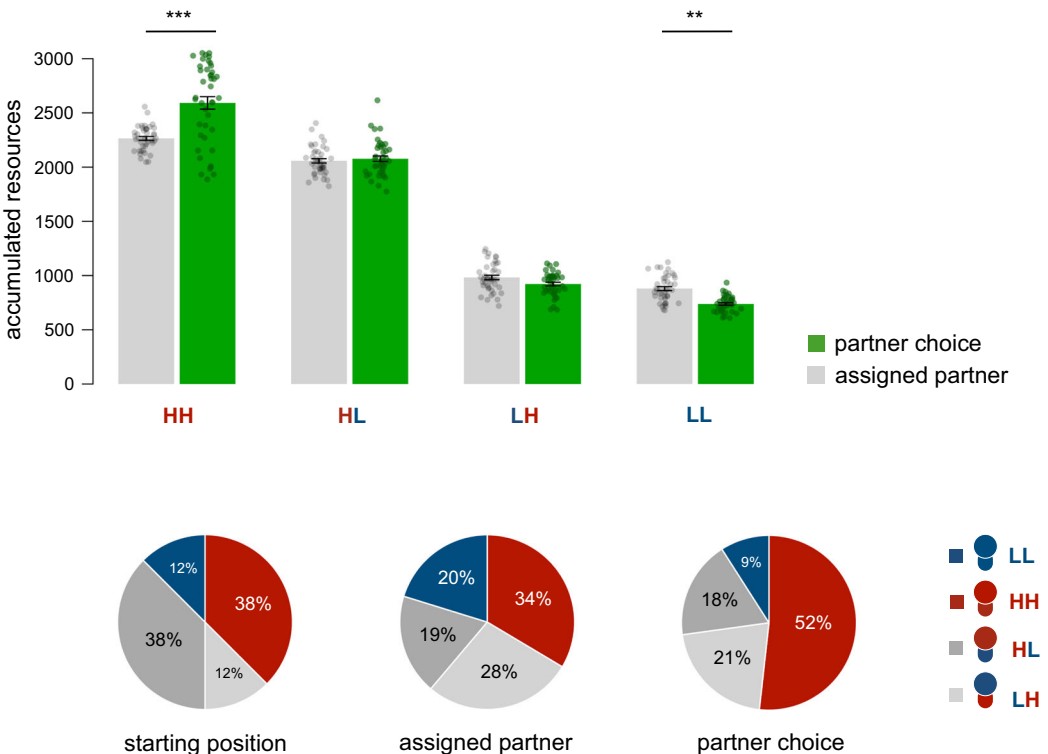

**Fig. 4 | Consequences of partner choice on cooperation and resource distribution. a** Cooperation (measured as average contributions to the public good as a percentage of participant's individual endowment) remained more stable in the partner choice condition (green line, $n = 21$ groups) compared to the assigned partner condition (black line, $n = 21$ groups; left panel). The right panel shows that participants differed in their cooperation rate towards partner types. While participants contributed more to the public good when paired with participants who were assigned a high endowment and high productivity factor (HH type), they cooperated less when paired with participants who were assigned a low endowment and low productivity factor (LL type) in the partner choice condition ($n = 21$ groups). The dotted line shows the average share of units invested in the public good in the partner choice condition (60.9%). **b** Partner choice increased resource disparity. Upper panel: accumulated resources per participant type (i.e., total number of units earned at the end of the experiment) in the partner choice (green bars, $n = 21$ groups) and assigned partner condition (grey bars, $n = 21$ groups). HH types benefitted from partner choice ($p < 0.001$), whereas LL types earned significantly less ($p = 0.005$) under partner choice compared to the assigned partner condition. Differences in accumulated resources were tested using a multilevel model (two-sided, Bonferroni corrected for multiple testing). Lower panel: Ex-ante resource distribution per type (left), received share of generated resources per type in the assigned partner condition (middle), and received share of generated resources in the partner choice condition (right) per type (blue = LL, red = HH, dark grey = HL, light grey = LH). Error bars and bands indicate the standard error of the mean. **\*\***$p < 0.01$, **\*\*\***$p < 0.001$. Dots show averages (**a**) and sums (**b**) per participant.

leads to a perpetuation of existing differences between individuals assigned a high endowment and high productivity factor, and individuals assigned a low endowment and low productivity factor. People prefer to be paired with partners assigned a high endowment and high productivity factor and cooperate with such partners to maximize welfare. As a result, individuals assigned a lower endowment and productivity factor become forced to work together, cooperate less well, and lack the capital or production facilities to render themselves attractive partners for cooperative exchange. Over time, this process of partner selection and cooperation helps those who are highly endowed and are assigned a high productivity factor to accumulate more resources, while prohibiting individuals who are assigned low endowment and low productivity factors from creating welfare.

Findings resonate with those of a recent study. In this study, Melamed and colleagues show how social network structures are affected by inequality[14]. Similar to our results, the study finds that people cooperate more with wealthier partners in order to maintain connections with them, thereby resulting in structural network change and producing greater system-level inequality. Interestingly, in this previous work, individuals could not directly reciprocate another person's actions, because they made a single decision vis-à-vis all those with whom they were connected. This means that participants could not exclusively give more to the advantaged, which is something our experimental design did allow for. As a result, we show that the segregation of individuals assigned a high endowment and high productivity factor versus the rest results from bidirectional preferences as well as stronger cooperation rates between these types when they chose to interact with each other.

While previous research showed how partner choice benefits cooperation when others could observe individuals' cooperation[10,11,14], we show how cooperation evolves in groups in which individuals were anonymous and in which they only were informed about others' general characteristics such as their endowment and productivity levels. Arguably, this emphasizes structural differences between participants and de-emphasizes cooperation behaviour when choosing partners. Our setting, therefore, captures situations in which people differ in some general characteristics (like endowment and productivity) that can be observed by others, while their behaviour (i.e., cooperation choices) can only be observed when interacting with them. In such situations, general characteristics may take precedence in partner choice and further fuel segregation and resource inequality, as we have also shown. An interesting question for future research would be to investigate how people integrate information if they have both knowledge on general characteristics (like endowment and productivity) and past behaviour (i.e., cooperation choices) when choosing a partner to cooperate with. We surmise that providing knowledge on others' past behaviour could create stronger concerns for direct reciprocity (since individuals become identifiable) and reveal the degree to which individual-level cooperativeness can compensate for structural disadvantages (like being endowed less or being less productive).

Another question for future research is whether current results depend on the size of the incentives used for the cooperation decisions. In the present experiment, incentives were calibrated to standards used in incentivized online studies, yet one may wonder whether current patterns generalize when decision-making has stronger financial consequences. While our data cannot answer that question, we note that meta-analyses provide some indication that behavior is often independent on the height, or even the presence, of incentives. For instance, while stake size can impact generosity in Dictator Games[30,31], stake size did not affect decision-making in Ultimatum Bargaining[30], and studies on the effect of in-group membership[22] and trust[32] on cooperation showed that incentivized decisions did not differ from hypothetical ones. Accordingly, we expect current findings to generalize to situations with stronger incentives.

In line with archival and econometric analyses[33,34] and resonating with the Matthew effect of accumulated advantage[35], we see that even in our experiment, using small 'artificial societies', the advantaged flock together, leading others to increasingly lag behind. Segregation or assortment, which can be enhanced by partner choice, but is also dependent on various other elements of social network structures, comes with increased cooperation within groups and defection between communities and neighborhoods[20,22,23]. Both segregation and wealth disparities have been linked to political polarization and violent conflict[36].

Whereas partner choice may enable individuals to build and maintain public goods from which everyone can benefit, we found that in artificially created societies in which individuals differed in endowment and productivity, partner choice can be a curse rather than cure: through partner selection, segregation endogenously emerges, and cooperation with similar others amplifies pre-existing differences between those who are advantaged and those who are not.

## Methods

### Participants and ethics
Our experiment was approved by the ethics committee of the Institute of Psychology at Leiden University (2020-11-12-M. Stallen-V2-2726) and pre-registered via AsPredicted (on March 12 2020; #53435, https://aspredicted.org/2NC_1V6). The experiment was programmed in oTree (version 3.4.0)[37] written in Python (version 3.7.9.). Participants were recruited using an online recruitment platform from Leiden University (The Netherlands, $n = 256$, 79% were female, self-reported gender) and using the online platform Prolific ($n = 80$, 39% were female, self-reported gender). For both platforms, we used identical incentive schemes, inclusion/exclusion criteria, and protocols. All data were collected online, and we collected an equal number of groups per condition via each platform (Leiden University: 32 groups in total, 16 groups per condition; Prolific: 10 groups in total, 5 groups per condition). No participants were excluded, and the experiment did not involve deception. Participants were between 18 and 48 years of age (M = 23.81, SD = 4.01), provided informed consent, and received full debriefing after participating. They received a standard fee of 8.15 euro for participation, and their decisions were incentivized (M = €0.96, SD = 0.28, range: 0.53–1.65€). Participation in the experiment took between 45 and 60 minutes.

### Main experiment
We used a 2 (condition: partner choice vs assigned partner) × 4 (type: High Endowment – High Productivity [HH], High Endowment – Low Productivity [HL], Low Endowment – High Productivity [LH], Low Endowment – Low Productivity [LL]) between-subjects design. Both conditions consisted of 21 groups of eight participants ($n = 168$ each). Each group consisted of two participants per type, so that there were 42 participants of each type per condition.

After giving informed consent, participants were instructed that their decisions, and those of other participants, would influence both their own payment and that of others. After the rules of the public goods game were explained, participants answered 13 practice questions to probe their understanding of the task. Only after all practice questions were answered correctly, participants could continue with the public goods game.

Participants played 24 rounds of a two-person, multiple-rounds public goods game. The public goods game confronts participants with a social dilemma. Every unit that is contributed to the public good is multiplied by a participant's productivity factor and is then evenly distributed among pairs. While mutually contributing all resources to the public good (i.e., full cooperation) increases joint welfare, it is always optimal to not cooperate and keep all units from a rational-selfish perspective. This is because (i) if a partner is not cooperating, it is best to not cooperate either, since the marginal return for each

invested unit is lower than keeping the unit for oneself (e.g., investing one unit to the public good with a productivity factor of 1.7 generates 1.7 units that are divided equally leading to a return of 0.85 units, which means cooperation generates a lower individual payoff compared to keeping the unit). If (ii) a partner is cooperating, participants can earn the most by withholding their own units, since, in this case, they do not pay the cost of cooperation but can free-ride on the cooperation of the partner. Hence, from a rational-economic (and selfish) perspective, regardless of what one's partner decides to do, it is in the best interest of the participant to not invest any resources into the public good in such a finitely repeated public goods game (assuming fixed partners).

Before the public goods game started, each participant was assigned an endowment and a productivity factor. The endowment factor represented the number of units each participant received at the start of each round of the public goods game, and was either 75 or 25 units. The productivity factor indicated with what number participant's contribution to the public good was multiplied, and was either 1.7 or 1.3. The combination of participants' endowment and productivity factor determined their type. In total, there were four types: the HH type (high endowment = 75, high productivity = 1.7), the HL type (high endowment = 75, low productivity = 1.3), the LH type (low endowment = 25, high productivity = 1.7), and the LL type (low endowment = 25, low productivity = 1.3). We used neutral labels to refer to these types during the experiment: type 1, type 2, type 3, and type 4.

Types were randomly assigned at the start of the experiment and fixed across the entire experiment. In the two-person public goods game, participants were paired with one fellow participant in their group (their partner) only identified by their type. After participants were introduced to the type of their partner for the current round, participants were asked to indicate how many units they wanted to contribute to the public good. At the end of each round, participants learned how many units their partner allocated to the public good and about their and their partner's earnings for this round before moving to the next round.

In the assigned partner condition, participants were, on each round, pseudo-randomly paired with a partner to play the public goods game with. Pairing was based on a computer algorithm that was programmed in Python (see Code Availability for availability of the pairing algorithm) such that participants played with each partner type six times, in a random order. In the partner choice condition, participants were asked, at the start of each round, to rank partner types based on their preference whom to play the public goods game with. Participants were paired with their first choice when there was another participant assigned a type of their first choice who placed the participant's type at the top of their list as well. For example, if a participant assigned an LL type preferred to be paired with an HH type, they were paired to this type if there was at least one other participant assigned an HH type who preferred to be paired with an LL type. When participants could not be paired with their first choice, because there were no other participants in their group who preferred to be paired with the type of the participant, the participant was paired with the partner type of their second choice. If this was not possible, they would be paired with their third choice, and so on. If two participants preferred to be paired with one available type, a random draw determined who got paired with the other participant. This matching algorithm was explained to participants before the start of the public goods game, so that all participants were aware of the mechanism underlying pair formation in the partner choice condition (see Code Availability for availability of the pairing algorithm). Before making their decision in the public goods game, participants learned to which type they were paired and then decided how many units they wanted to contribute to the public good, like in the assigned partner condition.

Before the start of the public goods game, participants were asked how many units they expected each type in their group to contribute on average when being paired with their type during the entire experiment. In the partner choice condition, participants were also asked once, at the start of the game, to indicate their expectations regarding which type participants expected to prefer to be paired with their own type.

After the public goods game, participants were asked to answer questions related to their strategies during the public goods game and completed a task to measure participants' social preferences, using the six-item social value orientation slider measure[38]. Furthermore, participants were asked to provide information about their demographics (age, gender, education, country, perceived socio-economic status[39], income, and the number of persons in their household). Finally, participants were informed about their earnings and were debriefed.

To determine serious participation, we included three attention checks and notified participants that failing two out of three attention checks would exclude them from data analysis (as pre-registered). All attentions checks were in capital letters and highlighted in yellow. First, after the practice questions, participants were asked to select the option Correct in response to the multiple-choice question: "Please select Correct with the options Correct and Incorrect". Second, after the final round of the public goods game, there was an attention check during which participants were asked to enter the number 200 in response to the question how many units they wanted to contribute to the public good. Finally, before the demographics questionnaire started, participants were asked to type the word green in a response box (spelling errors or differences in capital letters were not treated as missing this attention check). The first and third attention checks were answered correctly by all participants. Only the second attention check was missed by 20 participants. Following our pre-registered exclusion criterium, no participants were therefore excluded from the final analyses. We checked if our results would change after excluding the 20 participants who missed the second attention check; None of the reported results changed after exclusion.

Participants' decisions were incentivized by converting units to euros at the end of the experiment, with the conversion rate being 2600 units = €1.00. On average, participants in the partner choice condition earned 2670 units at the end of the experiment, and participants in the assigned partner condition earned 2481 units. These total earnings were based on the following incentives: Participants could earn units (1) in the public goods game, (2) by correctly guessing the cooperation rates of other participants, and (3) in the social value orientation task. Additionally, (4) in the partner choice condition, participants could earn units if they correctly guessed how other participants ranked their type. Regarding (1), the total number of units earned at the end of the experiment included the number of units earned across all public goods games rounds (accumulated resources), with an average of 1564.44 units across participants (SD = 725.13, range: 608–3052 units). Regarding (2), we compared participants' expectations with the actual, average, cooperation rate of each partner type. For each correct expectation, participants received 100 units, with a maximum of 400 units. Participants received, on average, an additional 22.62 units for correctly guessing the cooperation rates of their partners (SD = 49.04 units, range: 0–300 units). Regarding (3), to incentivize participants' decisions using the social value orientation slider measure, participants were randomly paired with another participant in their group. The choices of both pairs were incentivized, with each participant once being selected as the receiver and once as the allocator. On average, participants received an additional 912.88 units (SD = 56.65 units, range: 735–1041 units) for their choices in the social preference measure. Finally, in the partner choice condition, participants could earn additional units if they correctly guessed how other participants ranked the participant's type. For each correct expectation, participants received 100 units, with the maximum payment being 400 units per participant. On average, participants in the

partner condition, received 150.60 units (SD = 130.03 units, range: 0–400 units).

## Pre-registration

We pre-registered the experimental design, analysis plan, sample size, and exclusion criteria via AsPredicted (on March 12 2020; #53435, https://aspredicted.org/2NC_1V6). There were some deviations from our pre-registration. We planned to exclude participants when failing to respond in time twice, but we did not include maximum response time in the current public goods experiment. As a result we did not use this exclusion criterion. Furthermore, the first study in our pre-registration served as a pilot study to optimize the current experimental procedures (see Supplementary Note 2).

We included two main hypotheses in our pre-registration: (i) we hypothesized that most participants would prefer to be paired with an HH type (also HH types themselves). Thus, if non-HH types could not be paired with an HH type, we hypothesized that they would change their partner preference over time. This is in line with our findings. We also hypothesized that (ii) under partner choice, resource inequality (accumulated resources) would grow over time, which is in line with our findings as well.

## Statistical analyses

Models were implemented with the lme4 package in R[40]. If multiple contrasts were analysed within the same model, we corrected for multiple testing using a Bonferroni correction. All reported statistical tests were two-tailed. For all models reported we verified assumptions. Assumptions were met for most models. If assumptions were not met, we still used multilevel models, as these were most appropriate for our data structure and have been found to be robust to violations of distributional assumptions[41]. We also explored whether results differed between participant pools (32 groups recruited via Leiden University versus 10 groups recruited via Prolific) by computing, for each multilevel (logistic) model, an additional model with participant pool and all interaction terms including participant pool as covariates (see Supplementary Note 1).

**Segregation.** We fitted multilevel (logistic) models to investigate segregation under partner choice. Specifically, we investigated (i) how popular, (ii) how often rejected, and (iii) how often avoided HH and LL types were, as well as partner rankings and the stability of pairings over time. All models included random intercepts for participants nested within their group to account for violations of independence, since participants made repeated decisions and were part of a group in which they potentially influenced each other's decisions over time.

**Cooperation and resource distribution.** We fitted multilevel (logistic) models to investigate (i) how cooperation was impacted over time by the condition participants were in, (ii) if participants avoided uncooperative partner types, (iii) if differences in cooperation towards non-HH types depended on whether participants were an HH type themselves, (iv) if participants cooperated relatively more when their pairing was stable, (v) if differences in cooperation towards LL types depended on participants' own type, (vi) if differences in cooperation rate depended on whether participants were paired with their preferred partner type or not, and (vii) if accumulated resources (i.e., the total number of units at the end of the game) were impacted by condition and type. The multilevel model fitted to accumulated resources only included a random intercept for groups, since we analysed aggregated accumulated resources at the end of the public goods game per participant.

## Reporting summary

Further information on the research design is available in the Nature Portfolio Reporting Summary linked to this article.

## Data availability

The data of our experiment are publicly available in an OSF repository (https://doi.org/10.17605/OSF.IO/CASQZ)[42]. There are no restrictions to accessing the data. Additional information can be requested from the corresponding author at l.snijder@fsw.leidenuniv.nl.

## Code availability

The experiment and analysis code are publicly available in the same OSF repository (https://doi.org/10.17605/OSF.IO/CASQZ)[42].

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

## Acknowledgements
We thank R. Pliskin, A. Romano, L. Hoenig, and members of the Conflict and Cooperation lab at Leiden University for their feedback on an earlier version of this paper. This project has received funding from the European Research Council (ERC) under the European Union's Horizon 2020 research and innovation programme (AdG agreement n° 785635) to C.K.W.D.D., the Spinoza Award from the Netherlands Science Foundation (NWO SPI-57-242) to C.K.W.D.D., and a VENI Award from the Netherlands Science Foundation (NWO 016.Veni.195.078) to J.G.

## Author contributions
M.S., L.L.S., L.P.H., J.G. and C.K.W.D.D. conceived of the project and designed the studies. L.L.S. programmed the experiment and coordinated data collection. L.L.S. analyzed data with inputs from M.S. and J.G. M.S. and L.L.S. drafted the manuscript and incorporated co-author revisions.

## Competing interests
The authors declare no competing interests.
