## [Peer Review File · Nature Communications]

Partner choice and cooperation in social dilemmas can increase resource inequalityREVIEWER COMMENTS

Reviewer #1 (Remarks to the Author):

The authors seek to test whether partner choice exacerbates existing inequalities, because it allows individuals of high market value to selectively assort with each other, such that individuals with low market value are excluded from the gains of cooperation. This is an interesting question, given the importance of partner choice in human cooperation. The authors use a clever design for testing this question. There are some issues that should be resolved, but this manuscript has good potential.

Major Points

1) It sounds like participants couldn't stay with the same partner - only the partner type if there was partner choice - and couldn't keep a record of their partner's cooperation. This seems to downplay the effects of past cooperation, such that choice could only be based on ability (not willingness). This would likely exacerbate any effects of partner choice based on ability, given that choice based on individual willingness is not possible. This must be discussed as a limitation. This might also be why cooperation is maintained in the partner choice condition but declines in the assigned partner condition: there's more scope for repeated interactions with the same partner in the partner choice condition.

2) Earnings: Were all participants paid the same rates regardless of source (Leiden vs. Prolific)? The earnings are unclear. Are the public goods earnings included in the 2576 units (line 12 of methods), the 1564.44 units (line 15 of methods), or both? Are the 2576 units broken down into average earnings in each of the four components, or is this what the next four paragraphs are? (If so, then this needs to be clearer in writing).

If I understand the earnings correctly, with an exchange rate of 2600:1, it sounds like participants earned 1564.44 (~60 eurocents) for the public goods game, which is reasonable for Prolific, but seems low for in-person lab studies which typically take longer. Guessing others' cooperation resulted in an average of 22.62 units - at 2600:1, this is not even one euro cent. These stakes are so low as to be meaningless, even on Prolific. Then it's ~35 eurocents for social value orientation, and 5 eurocents for guessing other participants' rankings. These stakes would only be meaningful on Prolific, not in the lab.

3) Supplementary p5-6 Partner Choice: the authors must do a different second contrast, because these are not orthogonal. If HH is higher than the other three, then it increases the likelihood of LL being lower than the other three, because the "other three" in the latter comparison includes HH. If HH were much higher than the other three, but the other three were all identical, then both contrasts would be significant even if LL was the same as LH and HL. To fix this, the second contrast should be something like "LL vs. LH & HL", to see if LL really is lower than all the others, or whether it's just lower than HH.

There is a similar issue in Supplementary Table 3 & 4, and anywhere else one contrast is "HH vs the rest" and another contrast is "LL vs. the rest". This includes Supplementary Tables 11, 12, and possible others.

Minor Points

4) Were participant conditions counterbalanced across Leiden vs. Prolific? Presumably yes, but this should be explicitly stated.

5) In framing the hypotheses: Regarding patterns of partner preference, the "prefer high value partners hypothesis" (#3) doesn't just predict a preference for HH over all others, it also predicts that LL will be least chosen. HL and LH will be somewhere in the middle, with the exact preference

depending on whether endowment or multiplier has a bigger effect on one's returns with that kind of partner. It also predicts that all player types will have this preference, unless there is some cost associating with choosing (e.g., search cost, rejection cost), in which case players will aim for a partner who is "in their league" – if they're going to get someone the same as them anyway, they might as well avoid the search/rejection costs by aiming for someone of the same market value as themselves. See Barclay 2013 EHB 3.1.3 for details; this paper should be cited in this section of the results. Although there is no explicit cost of searching or rejection in this experiment, there may be a psychological cost of rejection, i.e., it's possible that many LL or LH people will just feel upset about never getting their first choice, such that they might cease ranking HH the highest to avoid the feeling of rejection. So "homophily" and "prefer high value partners" are not inherently in conflict – the former can result from the latter if people want to avoid being rejected. The hypotheses should be set up or discussed with reference to these points.

6) Figure 2a: is a spatial grid the best way of presenting this information? It's hard to get exact quantitative comparisons, as opposed to if this were presented as numbers instead a visual number of dots. Also, it's unclear why there would be (say) blue and red dots in the same block. For example, in the block that corresponds to LL with HH, is each dot meant to represent the number of HH partners that LL have, or the number of participants of either type in this pairing? I interpreted it as the former, which would suggest that they should all be red dots. It makes this presentation of the data unclear. I recommend switching to another format for presenting this information, unless the authors make it much clearer why this format is better than presenting it more numerically (e.g., bar graphs).

7) Figure 2c. Choices for HH decrease over time, which the authors interpret (rightfully in my opinion) as a decreasing preference for this type of partner and increasing homophily. However, there is an alternative explanation: if there is an HH player who defects, then experience with that particular HH participant makes people avoid that particular person, and since the experimental design doesn't allow them to avoid a particular person, they just avoid the type. Is there a way to tease apart these two explanations? If it were experience with a HH defector, then people with the most experience with HH will tend to decrease their HH choices. Conversely, if it's homophily as the LL & LH get sick of being rejected by HH, then the decrease in HH choices will come predominantly from those with the least experience with HH (i.e., low-value players); this is what the authors find. The increase in LL choice among LL players is also most consistent with homophily, rather than avoiding of particular HH players. So I think the evidence is most consistent with the authors' homophily explanation, but it's worth briefly discussing this somewhere, even if it's just mentioned briefly in main text and examine in slightly more detail in supplementary.

8) Figure 2d: this is presumably only in the partner choice condition, as these should all be equal in the assigned partner condition. Please state explicitly that this is the partner choice condition.

9) P9 last paragraph: can this be broken down by player type? Is the low cooperation with LL driven by higher-value types (e.g., HH), or is it even across the board – all types cooperate less with LL? The authors imply that the low cooperation with LL is driven by people not wanting to be with LL types – can they present a comparison of cooperation with LL when choosing LL and when not choosing LL? (And the same for other partner types). How does cooperation with a chosen HH compare with cooperation with a chosen LL? It would be good to have this info, even if just in Supplementary.

10) Figure 3a: can the red dots be made a darker red, and the blue dots a darker blue? This will make them easier to see against the background of the red and blue bars, respectively.

11) Discussion: the authors should mention that these results aren't specific to partner choice, but any type of assortment based on wealth/ability. For example, if societies interact entirely with kin (i.e., no partner choice beyond kin), but wealth or ability are associated with kinship, then this should produce the same results (i.e., higher inequality than if no such assortment). The authors' key argument is that partner choice is one kind of assortment, but they must be careful not to suggest that it's the

only assortment that will produce such results.

12) Type p 20 of supplementary, 2nd line: "parings" should be "pairings"

Reviewer #2 (Remarks to the Author):

This paper is very good. It does a phenomenal job of using an efficient and cleverly designed experiment to answer some important questions. In reading the paper, I had an experience that I'm not sure I've ever had before as a reviewer. In seven instances, I typed some version of "Yes, but what about this?" in the margins. The authors then did exactly what I was thinking they should do in the next paragraph or two. This happened for every one of my comments and concerns. It's a very strong paper.

I have one comment/suggestion. I would strongly encourage the authors to look at the following paper, just published in Scientific Reports

<https://www.nature.com/articles/s41598-022-10733-8>

Full disclosure: I am an author on this paper. As the authors will see, our paper addresses more or less the exact same issue as the paper I'm reviewing (including using the same key manipulations of player wealth and ability, though we use different labels, following the Hauser et al. paper cited by the authors). That said, the authors' paper (i.e., the paper I am reviewing) is much better executed than our paper. Their design is simpler, and the results are clearer (among other improvements) than ours. Further, the authors' design allows them to answer some questions that our design did not. Thus, the existence of our paper in Scientific Reports does not affect the magnitude of contribution the current paper makes whatsoever. I wish we'd written this paper rather than the one we published in Scientific Reports.

Reviewed by Brent Simpson

REVIEWER COMMENTS

Reviewer #1 (Remarks to the Author):

The authors seek to test whether partner choice exacerbates existing inequalities, because it allows individuals of high market value to selectively assort with each other, such that individuals with low market value are excluded from the gains of cooperation. This is an interesting question, given the importance of partner choice in human cooperation. The authors use a clever design for testing this question. There are some issues that should be resolved, but this manuscript has good potential.

Authors' Response: We would like to thank you for these constructive comments and your time. We carefully revised the manuscript in light of your comments and believe this improved the manuscript. In particular, we revised the main text, incorporated new analyses, and provided more methodological details. Below, we give a point-by-point response to all raised comments.

Major Points

1) It sounds like participants couldn't stay with the same partner - only the partner type if there was partner choice - and couldn't keep a record of their partner's cooperation. This seems to downplay the effects of past cooperation, such that choice could only be based on ability (not willingness). This would likely exacerbate any effects of partner choice based on ability, given that choice based on individual willingness is not possible. This must be discussed as a limitation. This might also be why cooperation is maintained in the partner choice condition but declines in the assigned partner condition: there's more scope for repeated interactions with the same partner in the partner choice condition.

Authors' Response: As you correctly point out, participants could not choose a specific participant to interact with, but participants chose with which *type* they preferred to (repeatedly) interact with, or not. Because we also did not provide any information about individual's past levels of cooperation, partner choice could thus indeed only be based on partner characteristics (ability and wealth) and not on direct knowledge on how a particular partner interacted in the past. This design choice was made because we specifically wanted to study how cooperation evolved in groups in which individuals were anonymous and in which they only were informed about others' wealth and ability levels. This mirrors, in our view, those situations in which people often cannot observe others' cooperativeness (except when interacting with them), and instead have to base their decision on general characteristics such as others' wealth and productivity levels.

As you mention, in the partner choice treatment, individuals could 'decipher' others' past cooperation rates when they were repeatedly paired to someone of their own type (e.g., HH paired to HH, or LH paired to LH). We thank you for raising this issue and now highlight in the Discussion that future research could be undertaken to experimentally control for having knowledge of others' past levels of cooperation, in addition to others' ability and wealth (page 13, line 21 - page 14, line 13):

While previous research showed how partner choice benefits cooperation when others could observe individuals' cooperation^{10,11,14}, we show how cooperation evolves in groups in which individuals were anonymous and in which they only were informed about others' general characteristics such as their wealth and productivity levels. Arguably, this emphasizes structural differences between participants and de-emphasizes cooperation behavior when choosing partners. Our setting, therefore, captures

situations in which people differ in some general characteristics (like wealth and ability) that can be observed by others, while their behavior (i.e., cooperation choices) can only be observed when interacting with them. In such situations, general characteristics may take precedence in partner choice and fuel segregation and inequality, as we have also shown. An interesting question for future research would be to investigate how people integrate information if they have both knowledge on general characteristics (like wealth and ability) and past behavior (i.e., cooperation choices) when choosing a partner to cooperate with. We surmise that providing knowledge on others' past behavior could create strong concerns for direct reciprocity (since individuals become identifiable) and reveal the degree to which individual-level cooperativeness can 'compensate' for structural disadvantages (like being less wealthy or able).

You correctly note that our design “...might also [explain] why cooperation is maintained in the partner choice condition but declines in the assigned partner condition: there's more scope for repeated interactions with the same partner in the partner choice condition.” Our prediction that especially HH types select other HH types to cooperate with results in repeated interactions between the two HH types (and as a consequence also among the two LL types) in our partner choice compared to our assigned partner treatment. We captured this finding in our measure of 'stability.' We find that stability correlates with level of cooperation, which fits extensive past work showing that past cooperation predicts future cooperation with that same partner, but this correlation between stability and cooperation is significantly weaker for LL types (Supplementary Information, page 26, lines 6-8):

However, the correlation between stability and cooperation is significantly weaker for LL types compared to HH types (multilevel model, $b_{\text{stability}} = -1.73$, $SE = 0.64$, $p = .007$, 95% CI [-2.98, -0.48], Supplementary Table 11).

Thus, we agree that one of the reasons that cooperation is maintained in the partner choice condition but declines in the assigned partner condition is that there's more scope for repeated interactions with the same partner in the partner choice condition.

2) Earnings: Were all participants paid the same rates regardless of source (Leiden vs. Prolific)? The earnings are unclear. Are the public goods earnings included in the 2576 units (line 12 of methods), the 1564.44 units (line 15 of methods), or both? Are the 2576 units be broken down into average earnings in each of the four components, or is this what the next four paragraphs are? (If so, then this needs to be clearer in writing).

If I understand the earnings correctly, with an exchange rate of 2600:1, it sounds like participants earned 1564.44 (~60 eurocents) for the public goods game, which is reasonable for Prolific, but seems low for in-person lab studies which typically take longer. Guessing others' cooperation resulted in an average of 22.62 units – at 2600:1, this is not even one euro cent. These stakes are so low as to be meaningless, even on Prolific. Then it's ~35 eurocents for social value orientation, and 5 eurocents for guessing other participants' rankings. These stakes would only be meaningful on Prolific, not in the lab.

Authors' Response: We apologize for not having made clearer that all data were collected online, and that only the online platform via which participants were recruited (Leiden University [32 groups] or Prolific [10 groups]) differed. For both platforms, we used identical incentive schemes, inclusion / exclusion criteria, and protocols.

We agree that the incentives for the public goods game, social value orientation measure, and for guessing other participants' rankings are reasonable for Prolific and adhere to common standards used in incentivized online studies. We also understand that our previous explanation of how much participants were paid across the different tasks was unclear, and we now clarify this in the Methods of the revised manuscript (page 20, lines 10-17):

On average, participants in the partner choice condition earned 2609 units at the end of the experiment, and participants in the assigned partner condition earned 2481 units. These total earnings were based on the following incentives: Participants could earn units (1) in the public goods game, (2) by correctly guessing the cooperation rates of other participants, and (3) in the social value orientation task. Additionally, (4) in the partner choice condition, participants could earn units if they correctly guessed how other participants ranked their type. See Supplementary Notes for how many units participants received on average per incentive. Participants' decisions were incentivized by converting units to euros at the end of the experiment, with the conversion rate being 2600 units = €1.00.

In the Supplementary Notes (page 2-3) we provide the following detail:

At the end of the experiment, participants in the partner choice condition owned 2609 units on average. Participants in the assigned partner condition owned 2481 units on average. These units, were, after converting to euros, paid as a bonus payment on top of a fixed show-up fee. Participants earned units in three (assigned partner condition) or four (partner choice condition) different ways, together summing up to the total number of units owned at the end of the experiment:

Firstly, the total number of units owned at the end of the experiment include the number of units that participants earned by all public goods games (accumulated wealth), with an average of 1564.44 units across participants (SD = 725.13, range: 608 – 3052 units).

Secondly, we compared participants' expectations with the actual, average, cooperation rate of each partner type. For each correct expectation, participants received 100 units, with a maximum of 400 units. Participants received, on average, an additional 18.45 units for correctly guessing the cooperation rates of their partners (SD = 45.17 units, range: 0 – 300 units).

Thirdly, to incentivize participants' decisions using the social value orientation slider measure, participants were randomly paired with another participant in their group. The choices of both pairs were incentivized, with each participant once being selected as the receiver and once as the allocator. On average, participants received an additional 912.88 units (SD = 56.65 units, range: 735 – 1041 units) for their choices in the social preference measure.

Finally, in the partner choice condition, participants could earn additional units if they correctly guessed how other participants ranked the participant's type. For each correct expectation, participants received 100 units, with the maximum payment being 400 units per participant. On average, participants in the partner condition, received 98.81 units (SD = 83.81 units, range: 0 – 300 units).

3) Supplementary p5-6 Partner Choice: the authors must do a different second contrast, because these are not orthogonal. If HH is higher than the other three, then it increases the likelihood of LL being lower than the other three, because the "other three" in the latter comparison includes HH. If HH were much higher than the other three, but the other three were all identical, then both contrasts would be significant even if LL was the same as LH and HL. To fix this, the second contrast

should be something like “LL vs. LH & HL”, to see if LL really is lower than all the others, or whether it’s just lower than HH.

There is a similar issue in Supplementary Table 3 & 4, and anywhere else one contrast is “HH vs the rest” and another contrast is “LL vs. the rest”. This includes Supplementary Tables 11, 12, and possible others.

Authors’ Response: We thank you for these suggestions for additional analyses. We gratefully used them and incorporated these new orthogonal contrasts to the relevant multilevel models comparing LL types with only LH and HL types. These new analyses complement and confirm our earlier conclusions, and we report them in the revised Supplementary Information. Also, we extended the explanation of the included contrasts (including all suggested contrasts) below each Supplementary Table. For your convenience, this is what we now report and conclude:

When comparing LL types with only LH and HL types, we find that:

- LL types are not preferred less (i.e., less popular) than HL and LH types; they are only significantly less popular than HH types.
 - Revised explanation in the Results of the manuscript (page 8, lines 1-4):
The high-endowment high-ability (HH) type was by far the most preferred partner type (Fig. 3a; i.e., most popular; 65.1% of all first choices of all types together consisted of an HH type; MLLM, $b_{type} = 5.56$, $SE = 0.57$, $p < 0.001$, Supplementary Table 3).
 - Supplementary Table 3 now includes a third contrast that tests whether LL types were less popular than HL and LH types. This difference is not significant, $b = -0.62$, $SE = 0.57$, $p = .78$.
 - These results are now discussed in the Supplementary Information (page 10, lines 8-13):
In contrast, LL types were the least preferred partner type (i.e., least popular; only 6.7% of all first choices of all types consisted of an LL type; multilevel logistic model, $b_{type} = -2.40$, $SE = 0.52$, $p < .001$, 95% CI [-3.59, -1.20], Supplementary Table 3), although preferences for an LL partner type did not differ significantly from preferences for HL and LH partner types (multilevel logistic model, $b_{type} = -0.62$, $SE = 0.54$, $p = .78$, 95% CI [-1.86, 0.63], Supplementary Table 3).
 - Revised explanation below Supplementary Table 3 (page 11):
Contrast 1 tests whether HH types were more often selected as participants’ first partner choice than the other types, and contrast 2 tests whether LL types were less often selected as participants’ first partner choice than the other types. Note that contrast 2 is not orthogonal to contrast 1. We therefore added contrast 3 which tests whether LL types were less often selected as participants’ first partner choice than HL and LH types.
- LL types are more often rejected than HL and LH types.
 - Supplementary Table 4 now includes a third contrast that tests whether LL types were more often rejected than HL and LH types. This difference is significant, $b = 0.78$, $SE = 0.31$, $p < .05$.
 - These results are now discussed in the Supplementary Information (page 12, lines 10-12):
LL types were also rejected more often than HL and LH types (multilevel logistic model, $b_{type} = 0.78$, $SE = 0.31$, $p < .05$, 95% CI [0.08, 1.48], Supplementary Table 4).
 - Revised explanation below Supplementary Table 4 (page 13):
Contrast 1 tests whether HH types could be paired with the type of their first choice more often than the other types, and contrast 2 tests whether LL types could be

paired with the type of their first choice less often than the other types. Note that contrast 2 is not orthogonal to contrast 1. We therefore added contrast 3 which tests whether LL types could be paired with the type of their first choice less often than HL and LH types could be paired with their first choice.

- Participants were more likely to switch partner preferences after being paired with an LL type, than after being paired with an HL and LH type.
 - Supplementary Table 5 now includes a third contrast that tests whether participants were more likely to switch partner preferences after being paired with an LL type than after being paired with HL and LH types. This difference is significant, $b = 0.48$, $SE = 0.13$, $p < .001$.
 - These results are now discussed in the Supplementary Information (page 14, lines 14-17):
Participants were also more likely to switch partner preferences after being paired with an LL type compared to after being paired with an HL and LH type (multilevel logistic model, $b_{\text{previous partner}} = 0.48$, $SE = 0.13$, $p < .001$, 95% CI [0.20, 0.77], Supplementary Table 5).
 - Revised explanation below Supplementary Table 5 (page 15):
Contrast 1 tests whether participants switched partner preferences less often after being paired with an HH type than after being paired with another type, and contrast 2 tests whether participants switched partner preferences more often after being paired with an LL type than after being paired with another type. Note that contrast 2 is not orthogonal to contrast 1. We therefore added contrast 3, which tests whether participants switched partner preferences more often after being paired with an LL type than after being paired with HL and LH types.
- Participants did not cooperate less with partners of their 4th preference than with partners of their 2nd and 3rd preference.
 - Revised explanation in the Results of the manuscript (page 10, lines 15-18):
Participants who were paired with the type of their first choice cooperated relatively more with their partner than those who were not paired with their first choice (MLM, $b_{\text{ranking}} = 14.11$, $SE = 1.06$, $p < 0.001$, Supplementary Table 13; controlling for own type, partner type, and round).
 - Supplementary Table 13 now includes a contrast that tests whether participants cooperated less with their last (4th) preference compared to their 2nd and 3rd preferences. This difference is not significant, $b = -2.96$, $SE = 1.54$, $p = .16$, indicating that participants' willingness to cooperate depends on (not) being paired to their first preference specifically.
 - These results are now discussed in the Supplementary Information (page 30, lines 16-19):
However, participants did not cooperate less with their last (fourth) preference compared to their second and third preference (multilevel model, $b_{\text{ranking}} = -2.96$, $SE = 1.54$, $p = .16$, 95% CI [-6.45, 0.52], Supplementary Table 13; controlling for own type, partner type, and round number).
 - Revised explanation below Supplementary Table 13 (page 31-32):
Contrast 1 tests whether participants cooperated more when being paired with their preferred partner type vs. when they were not paired with this type, and contrast 2 tests whether participants cooperated less when being paired with the partner type they least preferred vs. when they were not paired with this type. Note that contrast 2 is not orthogonal to contrast 1. We therefore added contrast 3 which tests whether participants cooperated more when being paired with their least preferred partner type vs. their second and third preference.

- In the partner choice condition, LL types accumulated significantly less wealth than HL and LH types.
 - Supplementary Table 14 now includes a sixth contrast that tests whether LL types accumulated less wealth in the partner choice condition compared to HL and LH types. This difference is significant, $b = -761.39$, $SE = 29.08$, $p < .001$.
 - These results are now discussed in the Supplementary Information (page 34, lines 4-6):
Finally, LL types also accumulated less wealth than HL and LH types in the partner choice condition (multilevel model, $b_{type} = -761.39$, $SE = 29.08$, $p < .001$, 95% CI [-835.27, -687.51], Supplementary Table 14).
 - Revised explanation below Supplementary Table 15 (page 35-36):
Contrast 1 tests whether participants accumulated more wealth in the partner choice condition than in the assigned partner condition, contrast 2 tests whether HH types accumulated more wealth in the partner choice condition than in the assigned partner condition, contrast 3 tests whether LL types accumulated less wealth in the partner choice condition than in the assigned partner condition, contrast 4 tests whether HH types accumulated more wealth in the partner choice condition than the other types did, and contrast 5 tests whether LL types accumulated less wealth in the partner choice condition than the other types did. Note that contrast 5 is not orthogonal to contrast 4. We therefore added contrast 6 which tests whether LL types accumulated less wealth in the partner choice condition than was accumulated by HL and LH types.

Minor Points

4) Were participant conditions counterbalanced across Leiden vs. Prolific? Presumably yes, but this should be explicitly stated.

Authors' Response: As we now clarified in the main text, we collected an equal number of groups per condition on each platform (Leiden: 32 groups in total, 16 groups per condition; Prolific: 10 groups in total, 5 groups per condition).

Due to the COVID pandemic it was not possible to recruit more participants using our Leiden University participant pool. Therefore, we collected 10 groups on Prolific using identical protocols, inclusion / exclusion criteria, and incentive schemes. We now explicitly mention this in the Methods of the revised manuscript (page 16, line 5-11):

Participants were recruited using an online recruitment platform from Leiden University (The Netherlands, $N = 256$, 79% were female, self-reported gender) and using the online platform Prolific ($N = 80$, 39% were female, self-reported gender). For both platforms, we used identical incentive schemes, inclusion / exclusion criteria, and protocols. All data were collected online, and we collected an equal number of groups per condition via each platform (Leiden University: 32 groups in total, 16 groups per condition; Prolific: 10 groups in total, 5 groups per condition).

5) In framing the hypotheses: Regarding patterns of partner preference, the “prefer high value partners hypothesis” (#3) doesn't just predict a preference for HH over all others, it also predicts that LL will be least chosen. HL and LH will be somewhere in the middle, with the exact preference depending on whether endowment or multiplier has a bigger effect on one's returns with that kind of partner. It also predicts that all player types will have this preference, unless there is some cost

associating with choosing (e.g., search cost, rejection cost), in which case players will aim for a partner who is “in their league” – if they’re going to get someone the same as them anyway, they might as well avoid the search/rejection costs by aiming for someone of the same market value as themselves. See Barclay 2013 EHB 3.1.3 for details; this paper should be cited in this section of the results. Although there is no explicit cost of searching or rejection in this experiment, there may be a psychological cost of rejection, i.e., it’s possible that many LL or LH people will just feel upset about never getting their first choice, such that they might cease ranking HH the highest to avoid the feeling of rejection. So “homophily” and “prefer high value partners” are not inherently in conflict – the former can result from the latter if people want to avoid being rejected. The hypotheses should be set up or discussed with reference to these points.

Authors’ Response: We thank you for highlighting this – it is also in full agreement with how we understand (and anticipated) our results. We have made more explicit that our third hypothesis not only predicts a preference for HH types, but that it predicts that partners with an LL type are preferred least as well (Introduction, page 5, lines 21-23):

The third possible pattern of results is based on the idea that people in general prefer wealthy-and-able partners (HH types) while not preferring poor-and-unable partners (LL types), as cooperating with the former more likely generates value to the individual^{7,10,14}.

Furthermore, when discussing our second hypothesis on homophily preferences, we now highlight how this could be driven by (psychological) rejection costs and can result from participants preferring high value partners. We also cite Barclay 2013 as it indeed is quite fitting (Introduction, page 5, lines, 12-14):

Such homophily preferences may be grounded in empathy avoidance²⁴⁻²⁶ or in avoiding the (psychological) costs of rejection by wealthier and/or more able partners⁶.

We also return to this issue in our Results section (page 8, line 22 - page 9, line 3):

However, the preference for HH types decreased over time for participants with a different type (MLLM, $b_{round} = -0.07$, $SE = 0.01$, $p < 0.001$, Supplementary Table 6), probably because HH types consistently preferred to be paired with each other (MLLM, $b_{round} = 0.002$, $SE = 0.02$, $p = 0.88$, Supplementary Table 6) and were out of reach for the other types, or because non-HH types wanted to avoid rejection.

And we adjusted the Discussion (page 13, lines 5-8):

People prefer to be paired with wealthy-and-able partners and cooperate with such partners to maximize welfare. As a result, less wealthy-and-able individuals become ‘forced’ to work together, cooperate less well, and lack the capital or skills to render themselves attractive partners for cooperative exchange.

Regarding your last point: We totally agree that homophily preferences can result from trying to reduce rejection rates (which is learned over time). Along this point, we write in the Results (page 9, lines 4-6):

This suggests that partner choice can create homophily preferences over time, with rejection avoidance being a possible driver of this effect. Importantly, homophily emerged as a consequence rather than cause of population segregation.

6) Figure 2a: is a spatial grid the best way of presenting this information? It's hard to get exact quantitative comparisons, as opposed to if this were presented as numbers instead a visual number of dots. Also, it's unclear why there would be (say) blue and red dots in the same block. For example, in the block that corresponds to LL with HH, is each dot meant to represent the number of HH partners that LL have, or the number of participants of either type in this pairing? I interpreted it as the former, which would suggest that they should all be red dots. It makes this presentation of the data unclear. I recommend switching to another format for presenting this information, unless the authors make it much clearer why this format is better than presenting it more numerically (e.g., bar graphs).

Authors' Response: We added a new figure in which the data of Fig. 2a is presented in a stacked bar graph (Fig. 2b, page 7). Since Fig. 2a presents the raw data, we did not replace this figure, but we clarified the figure caption (page 6, lines 19 – 27):

a Spatial grid displaying the frequency of observed types in each possible pairing configuration in the assigned partner (left) and partner choice condition (right). Each dot represents one observation per type and pairing. For example, the top right block which corresponds to a pairing configuration of an LL type interacting with an HH type: The blue dots represent the number of LL types, and the red dots represent the number of HH types that were in this pairing configuration in the assigned partner condition or in the partner choice condition. All dots together reflect the number of participants that were part of a HH-LL pair. The frequency of dots along the diagonal shows that partner choice led to a segregation of the population into pairs of similar types (higher frequency of dots on the diagonal in the partner choice condition than in the assigned partner condition).

For the revision, we decided to keep the spatial grid, since it allows to visually and quickly capture the large degree of same-type pairings (as illustrated along the diagonal compared to the off-diagonal). This can also be seen in the bar plot, but it is less obvious from first sight, since one has to compare each bar and corresponding type-colour. The bar plot, on the other hand, allows to derive the exact numbers/frequencies, as you point out.

7) Figure 2c. Choices for HH decrease over time, which the authors interpret (rightfully in my opinion) as a decreasing preference for this type of partner and increasing homophily. However, there is an alternative explanation: if there is an HH player who defects, then experience with that particular HH participant makes people avoid that particular person, and since the experimental design doesn't allow them to avoid a particular person, they just avoid the type. Is there a way to tease apart these two explanations? If it were experience with a HH defector, then people with the most experience with HH will tend to decrease their HH choices. Conversely, if it's homophily as the LL & LH get sick of being rejected by HH, then the decrease in HH choices will come predominantly from those with the least experience with HH (i.e., low-value players); this is what the authors find. The increase in LL choice among LL players is also most consistent with homophily, rather than avoiding of particular HH players. So I think the evidence is most consistent with the authors' homophily explanation, but it's worth briefly discussing this somewhere, even if it's just mentioned briefly in main text and examine in slightly more detail in supplementary.

Authors' Response: We thank you for this suggestion. To address this point, we ran an additional multilevel regression model to test whether the decreasing preference for HH types was driven by non-HH types avoiding HH type defectors. Results suggest that this alternative explanation could have played a role, in addition to an increase in homophily. Specifically, the model shows that non-

HH types experience relatively more partner-defection (i.e., less cooperation) when paired with HH types compared to when paired with other types. This increase in defection could have motivated the non-HH types to avoid HH types and move to, e.g., choosing their own type more often. We report this analysis in the Supplementary Information (page 24, lines 1-11; Supplementary Table 10):

We fit a multilevel regression model to investigate if, in addition to an increase in homophily, the decreasing preference for HH types could be driven by non-HH types avoiding HH type defectors. If so, participants with a non-HH type should have met more cooperative non-HH types than cooperative HH types. The model therefore only included participants with a non-HH type. The dependent variable was the relative cooperation of one's partner. Fixed effects were a dummy variable coding whether participants were paired with an HH type (= TRUE), or not (= FALSE), round number, and their interaction.

HH partners cooperated significantly less than non-HH partners when being paired with a non-HH type (multilevel model, $b_{partner} = -22.52$, $SE = 3.76$, $p < .001$, 95% CI [-29.88, -15.16], Supplementary Table 10), suggesting that the decrease in preference for HH types could (also) be driven by non-HH types avoiding HH type defectors.

We refer to these findings in the Results of the revised manuscript (page 10, lines 4-7):

Indeed, results show that participants changed their partner preference if, on the previous round, their partner cooperated relatively less than they did (MLLM, $b_{contribution} = 0.34$, $SE=0.10$, $p<0.001$, Supplementary Table 9). This could also explain why the preference of non-HH types for HH types decreased over time (Supplementary Table 10).

8) Figure 2d: this is presumably only in the partner choice condition, as these should all be equal in the assigned partner condition. Please state explicitly that this is the partner choice condition.

Authors' Response: You correctly assumed that the data reported in Fig. 2d only applied to the partner choice condition. We now changed Fig. 2d into Fig. 2c and adjusted the figure caption accordingly (page 7, lines 3-6):

c. Average length of consecutive interactions between different pairs (as a measure of pair stability) in the partner choice condition. Whereas HH-HH pairs interacted for an average of 9.7 consecutive rounds, HH-LL pairs were least stable and only interacted for an average of 1.4 consecutive rounds before breaking up. Error bars indicate the standard error of the mean.

9) P9 last paragraph: can this be broken down by player type? Is the low cooperation with LL driven by higher-value types (e.g., HH), or is it even across the board – all types cooperate less with LL? The authors imply that the low cooperation with LL is driven by people not wanting to be with LL types – can they present a comparison of cooperation with LL when choosing LL and when not choosing LL? (And the same for other partner types). How does cooperation with a chosen HH compare with cooperation with a chosen LL? It would be good to have this info, even if just in Supplementary.

Authors' Response: We thank you for these suggestions. We ran additional models to address these questions.

We now show the cooperation rate towards all partner types, broken down per player type, in Supplementary Fig. 1 (page 4-5):

Supplementary Fig. 1 shows the relative cooperation towards partner types, depicted separately for participants' own type. As can be seen, HH types cooperate less when being paired with LH or LL types than when being paired with HH or HL types. However, the cooperation of LL types is not dependent on their partner type.

We also ran an additional model to investigate if the low cooperation towards LL types depends on higher value types (i.e., HH, HL, and LH types) reducing their cooperation towards LL types. We report our findings in the Supplementary Information and refer to these findings in the Results of the revised manuscript.

Supplementary Information (page 28, lines 2-12; Supplementary Table 12):

We fit a multilevel regression model to investigate if the low cooperation towards LL types depended on higher value types (i.e., HH, HL, and LH types) reducing their cooperation towards LL types, or not. Therefore, we included LL types as the reference level in the model. The dependent variable was participant's relative cooperation rate. The fixed effects were a dummy variable coding whether participants were paired with an LL type (= TRUE) or not (= FALSE), participant's own type, and the interaction between these two variables.

Results showed that HH, HL, and LH types reduced their cooperation rates significantly when being paired with an LL partner compared to when being paired with a non-LL partner (multilevel model, $b_{LLpartner \times HHtype} = -9.99$, $SE = 4.58$, $p < .05$, 95% CI [-18.99, -1.05]; $b_{LLpartner \times HLtype} = -18.51$, $SE = 2.97$, $p < .001$, 95% CI [-24.34, -12.72], $b_{LLpartner \times LHtype} = -18.69$, $SE = 2.86$, $p < .001$, 95% CI [-24.29, -13.10], Supplementary Table 12).

Manuscript (page 10, lines 11-12):

While, overall, partner choice prevented the breakdown of cooperation, cooperation levels depended on partner type (Fig. 4a, see also Supplementary Fig. 1 and Supplementary Table 12).

In addition, you ask why we imply that low cooperation with an LL type is driven by people not wanting to be with LL types. We actually did not mean to imply this, and believe this misunderstanding originated from us testing whether participants' cooperation rate depended on whether they were paired with their first choice or not. We report the main effect of this analysis in the Results of the manuscript (i.e., applying to all types, controlling for partner type, and round; page 10, lines 16-18). However, to answer your question, we would need to perform three-way interactions between partner preferences, participant's own type and their partner's type. We refrained from doing this, because the number of observations needed for this analysis was severely unbalanced in our design (i.e. high types rarely chose to be paired with LL types, and actually were rarely paired with LL types compared to other types), making the results of this analysis unreliable.

Lastly, you ask whether cooperation with one's first choice depends on this first choice being either an HH type or an LL type. We believe this is an interesting question, but also for this analysis the number of observations in our design is very unbalanced (i.e., HH types were much more often preferred to be paired with than LL types), and therefore are not added to the revised manuscript.

We now clarified the results in the manuscript that describe whether cooperation depended on participants being paired with their first choice or not (page 10, lines 15-18):

Participants who were paired with the type of their first choice cooperated relatively more with their partner than those who were not paired with their first choice (MLM, $b_{ranking} = 14.11$, $SE = 1.06$, $p < 0.001$, Supplementary Table 13; controlling for own type, partner type, and round).

In the Results section in the manuscript, we also are now more precise in how we explain these results (page 10, lines 14-15):

These differences were driven by whether participants were paired with their preferred partner type or not.

10) Figure 3a: can the red dots be made a darker red, and the blue dots a darker blue? This will make them easier to see against the background of the red and blue bars, respectively.

Authors' Response: In the revised manuscript, Figure 3a is now Figure 4a (page 11). We made the bars transparent, such that the dots can be easily observed.

11) Discussion: the authors should mention that these results aren't specific to partner choice, but any type of assortment based on wealth/ability. For example, if societies interact entirely with kin (i.e., no partner choice beyond kin), but wealth or ability are associated with kinship, then this should produce the same results (i.e., higher inequality than if no such assortment). The authors' key argument is that partner choice is one kind of assortment, but they must be careful not to suggest that it's the only assortment that will produce such results.

Authors' Response: We have now explained in our Discussion that our results are not specific to partner choice, but that segregation can also depend on various other elements of social network structures (page 14, line 23 – page 15, line 4):

Segregation or assortment, which can be enhanced by partner choice, but is also dependent on various other elements of social network structures, comes with increased cooperation within groups and defection between communities and neighborhoods^{20,22,23}, and both segregation and wealth disparities have been linked to political polarization and violent conflict³⁵.

12) Type p 20 of supplementary, 2nd line: "parings" should be "pairings"

Authors' Response: We thank you for catching this typo and corrected it accordingly.

Reviewer #2 (Remarks to the Author):

This paper is very good. It does a phenomenal job of using an efficient and cleverly designed experiment to answer some important questions. In reading the paper, I had an experience that I'm not sure I've ever had before as a reviewer. In seven instances, I typed some version of "Yes, but what about this?" in the margins. The authors then did exactly what I was thinking they should do in the next paragraph or two. This happened for every one of my comments and concerns. It's a very strong paper.

I have one comment/suggestion. I would strongly encourage the authors to look at the following paper, just published in Scientific Reports

<https://www.nature.com/articles/s41598-022-10733-8>

Full disclosure: I am an author on this paper. As the authors will see, our paper addresses more or less the exact same issue as the paper I'm reviewing (including using the same key manipulations of player wealth and ability, though we use different labels, following the Hauser et al. paper cited by the authors). That said, the authors' paper (i.e., the paper I am reviewing) is much better executed than our paper. Their design is simpler, and the results are clearer (among other improvements) than ours. Further, the authors' design allows them to answer some questions that our design did not. Thus, the existence of our paper in Scientific Reports does not affect the magnitude of contribution the current paper makes whatsoever. I wish we'd written this paper rather than the one we published in Scientific Reports.

Reviewed by Brent Simpson

Authors' Response: We thank you very much for your kind words. We have to admit that we were not aware of this paper at the time of submission. We believe that the study reported in Scientific Reports is very interesting, and we believe that both studies nicely complement our understanding of how a priori inequality and partner choice can influence cooperation and a posteriori inequality.

We are happy to now have the opportunity to refer to this publication in our manuscript and explain how our experimental design allows us to answer several important questions that complement the insights from the paper in Scientific Reports.

In the Introduction, we now included multiple references to this paper¹⁴.

Page 2, lines 13-14:

Past experimental work on partner choice in social dilemmas typically assumed that individuals have the same ability to reciprocate cooperation (but see^{13,14}).

Page 2 lines 20-21:

Recent work started to address how inequality affects cooperation in social networks¹⁴ but how partner choice affects cooperation in social dilemmas among unequals remains an open question.

Page 5, lines 21-23:

The third possible pattern of results is based on the idea that people in general prefer wealthy-and-able partners (HH types) while not preferring poor-and-unable partners (LL types), as cooperating with the former more likely generates value to the individual^{7,10,14}.

We also more extensively discuss this publication in the Discussion (page 13, lines 11-20):

Findings resonate with those of a recent study. In this study, Melamed and colleagues show how social network structures are affected by inequality¹⁴. Similar to our results, the study finds that people cooperate more with wealthier partners in order to maintain connections with them, thereby resulting in structural network change and producing greater system-level inequality. Interestingly, in this previous work individuals could not directly reciprocate another person's actions, because they made a single decision vis-à-vis all those with whom they were connected. This means that participants could not exclusively give more to the wealthy, which is something our experimental design did allow for. As a result, we show that the segregation of populations in wealthy-and-able versus the rest results also from bidirectional preferences for the wealthy-and-able as well as stronger cooperation rates between these types when they found each other.

REVIEWER COMMENTS

Reviewer #1 (Remarks to the Author):

The authors have addressed the majority of my comments. However, I still have a concern about the small stake sizes. Stakes were minimal: the public goods game appears to have had 1 cent endowments per round for participants with low wealth, 3 cents for participants with high endowment. Even across all rounds, it works out to about 60 eurocents per participant. This is so small as to be practically hypothetical, especially in the lab. (The other tasks also had small stakes.) At the very least, this requires serious discussion to justify – more than just a quick sentence. How are these stakes meaningful enough to be better than hypothetical decisions? Or if they're not, why is it OK to use hypothetical decisions like this? To what extent would the results generalize to larger stakes? Normally, monetary incentives help to overcome demand characteristics (i.e., giving socially appropriate responses or what they think the experiments "want to hear"), but these monetary incentives are so small that they might not overcome demand characteristics – are there any demand characteristics that we need to worry about? Is there a chance that tiny stakes are actually worse than hypothetical decisions – is it like that classic daycare study in behavioural economics where fines made people arrive later, or the general conclusion that one should "pay enough or don't pay at all"? These are some of the questions that come up for me when I see stakes this small. The authors must do more to address the fact that their stakes were very small.

One solution is to compare effect sizes in the lab vs. Prolific. These stakes aren't as bad on Prolific as they are in the lab, so if the effect sizes are comparable in the lab as on Prolific, then this helps defend against the attack that the stakes are too small to be meaningful. Obviously such an analysis would be underpowered, and wouldn't reach statistical significance – it's more of an informal comparison of effect sizes to see whether effect sizes are noticeably larger in the lab (where demand characteristics would be more prevalent) than on Prolific (where demand characteristics are smaller in part before the stakes are more appropriate). If effect sizes are in the same ballpark, then that would be reassuring. This analysis could be in supplementary material and briefly mentioned in main text. This analysis is preferable to a verbal argument on the small stake size, because it would back up the claim with data.

Minor recurrent spelling error: Search for all occurrences of "paring" or "parings" and change to "pairing(s)", as some mistakes remain. Do this for both main text and supplementary.

Reviewer #2 (Remarks to the Author):

no additional comments or suggestions.

REVIEWER COMMENTS

Reviewer #1 (Remarks to the Author):

The authors have addressed the majority of my comments. However, I still have a concern about the small stake sizes. Stakes were minimal: the public goods game appears to have had 1 cent endowments per round for participants with low wealth, 3 cents for participants with high endowment. Even across all rounds, it works out to about 60 eurocents per participant. This is so small as to be practically hypothetical, especially in the lab. (The other tasks also had small stakes.) At the very least, this requires serious discussion to justify – more than just a quick sentence. How are these stakes meaningful enough to be better than hypothetical decisions? Or if they're not, why is it OK to use hypothetical decisions like this? To what extent would the results generalize to larger stakes? Normally, monetary incentives help to overcome demand characteristics (i.e., giving socially appropriate responses or what they think the experiments “want to hear”), but these monetary incentives are so small that they might not overcome demand characteristics – are there any demand characteristics that we need to worry about? Is there a chance that tiny stakes are actually worse than hypothetical decisions – is it like that classic daycare study in behavioural economics where fines made people arrive later, or the general conclusion that one should “pay enough or don't pay at all”? These are some of the questions that come up for me when I see stakes this small. The authors must do more to address the fact that their stakes were very small.

One solution is to compare effect sizes in the lab vs. Prolific. These stakes aren't as bad on Prolific as they are in the lab, so if the effect sizes are comparable in the lab as on Prolific, then this helps defend against the attack that the stakes are too small to be meaningful. Obviously such an analysis would be underpowered, and wouldn't reach statistical significance – it's more of an informal comparison of effect sizes to see whether effect sizes are noticeably larger in the lab (where demand characteristics would be more prevalent) than on Prolific (where demand characteristics are smaller in part before the stakes are more appropriate). If effect sizes are in the same ballpark, then that would be reassuring. This analysis could be in supplementary material and briefly mentioned in main text. This analysis is preferable to a verbal argument on the small stake size, because it would back up the claim with data.

Minor recurrent spelling error: Search for all occurrences of "paring" or "parings" and change to "pairing(s)", as some mistakes remain. Do this for both main text and supplementary.

Authors' Response: We would again like to thank you for your comments and your time. We revised the manuscript in light of your comments and believe this improved the manuscript.

We would like to clarify that all data were collected online and no data was collected in the lab. Only the online platform via which participants were recruited (Leiden University platform or Prolific platform) differed between studies. On both platforms, we collected data online with identical incentive schemes, inclusion / exclusion criteria, and protocols (see Methods; page 16, lines 6 – 11).

Since all data were collected online, the analyses that you suggest cannot tell us whether results differ between on-line and in-lab participation. Instead, they would tell us whether the results in our study differ between the Prolific sample and those from our Leiden subject pool. We nevertheless performed these analyses and found that results do not differ between platforms. Specifically, when we included “sample type” as covariate in our analyses, no effects for the covariate emerge

anywhere and the effects reported in the manuscript and on-line SI remain. We now explicitly mention that results are similar for both platforms in the Methods (page 16, lines 11 – 12):

Results did not differ between platforms.

We do agree, however, that it is an open question whether the current results depend on the stake size used and we now elaborate on this in more detail in the Discussion (page 14, lines 12 – 21):

Another question for future research is whether current results depend on the size of the incentives used for the cooperation decisions. In the present experiment, incentives were calibrated on common standards used in incentivized online studies, yet one may wonder whether current patterns generalize when decision-making has stronger financial consequences. While our data cannot answer that question, we note that meta-analyses show that behaviour is often independent on the height, or even the presence, of incentives. For instance, while evidence is mixed for generosity in Dictator Games³⁰⁻³², stake size does not affect decision-making in Ultimatum Bargaining³⁰, and studies on the effect of in-group membership²² and trust³³ on cooperation showed no difference between hypothetical and incentivized decisions. Accordingly, we expect current findings to generalize to situations with larger incentives.

We also added with which software the experiment was programmed in the Methods (page 16, line 5):

The experiment was programmed in oTree (version 3.4.0)⁴⁰.

We hope that our response resolves your remaining concerns. We again like to thank you for your valuable feedback and are grateful for spotting some remaining typos as well.

REVIEWERS' COMMENTS

Reviewer #1 (Remarks to the Author):

The authors have addressed my concerns satisfactorily. I am happy to recommend publication with some very minor changes. I do not need to see another revision of the manuscript – I am happy for the editor to confirm these.

First, given the small incentives and potential concern about this, it is worth briefly mentioning the average time length of the experiment, so readers can assess whether 96 Eurocents of incentives is indeed in line with common standards for incentives in other online experiments for an experiment of that length. (The minimum base payment of 8.15 Euro is in line with common standards.)

Second, thank you for running a comparison of the Prolific vs. Leiden participants. However, simply including the sample as a covariate would not detect anything other than a main effect of the sample on the DV (e.g., lower contributions in one sample), which is not the question. If effects were bigger in one sample, such as partner choice affecting inequality on Prolific but not Leiden, then this would be appear as an interaction between sample and any of the other effects. The authors should run this analysis to test for interactions, to ensure that the effects are not bigger in the Prolific sample (who may be used to smaller incentives) than the Leiden sample. I am assuming that the effects are the same, and there are no notable interactions, so I am not demanding to see another revision of the manuscript before publication. However, if the effects *are* different (i.e., if there is an interaction between the sample type and any other effects), then the authors must note these in the final manuscript. They should confirm to the editor whether there are any such interactions.

Third, a really nit-picking point, but the evidence for stake size in Dictator Games is not as mixed as the authors claim, given that two meta-analyses show a small effect of stakes, and that one of the two “no effect” papers the authors cite is already included in those meta-analyses. (The other is too new for those meta-analyses, and has its own problems for testing this question, like only paying some participants, which reduces the stakes and thus the effect size of stakes.) So this should be toned down rather than simply saying the results are mixed – they results clearly lean towards one side. Further, the authors should cite the Engel 2011 meta-regression (Experimental Economics), which combines many, many Dictator Game studies to show less generosity at higher stakes. This would represent a more accurate picture of the literature on stakes in Dictator Games.

REVIEWER COMMENTS

Reviewer #1 (Remarks to the Author):

The authors have addressed my concerns satisfactorily. I am happy to recommend publication with some very minor changes. I do not need to see another revision of the manuscript – I am happy for the editor to confirm these.

Authors' Response: We again thank you for your time and helpful suggestions. We are happy to hear that we could address your previous concerns satisfactorily. We now revised the manuscript in response to your remaining comments, and made some additional changes in response to those of the editorial team. We believe the current manuscript improved substantially based on your valuable feedback. Thank you!

First, given the small incentives and potential concern about this, it is worth briefly mentioning the average time length of the experiment, so readers can assess whether 96 Eurocents of incentives is indeed in line with common standards for incentives in other online experiments for an experiment of that length. (The minimum base payment of 8.15 Euro is in line with common standards.)

Authors' Response: We now report the duration of the experiment in the Methods (page 14, line 16):

Participation in the experiment took between 45 and 60 minutes.

Second, thank you for running a comparison of the Prolific vs. Leiden participants. However, simply including the sample as a covariate would not detect anything other than a main effect of the sample on the DV (e.g., lower contributions in one sample), which is not the question. If effects were bigger in one sample, such as partner choice affecting inequality on Prolific but not Leiden, then this would appear as an interaction between sample and any of the other effects. The authors should run this analysis to test for interactions, to ensure that the effects are not bigger in the Prolific sample (who may be used to smaller incentives) than the Leiden sample. I am assuming that the effects are the same, and there are no notable interactions, so I am not demanding to see another revision of the manuscript before publication. However, if the effects *are* different (i.e., if there is an interaction between the sample type and any other effects), then the authors must note these in the final manuscript. They should confirm to the editor whether there are any such interactions.

Authors' Response: We now conducted, for each multilevel (logistic) model reported in the manuscript and in the Supplementary Information, an additional model with participant pool as a covariate and interaction terms for any possible interactions between participant pool and any of the reported effects. Because we had no hypotheses, and in total 59 interaction terms involving participant pool were tested, we took care to avoid Type I error and applied Bonferroni correction on the significance threshold. This resulted in four reliable interaction effects. Analyses are explained in full in the SI, and are succinctly described in the Main Text (page 20, lines 7-10):

We also explored whether results differed between participant pools (32 groups recruited via Leiden University versus 10 groups recruited via Prolific) by computing, for each multilevel (logistic) model, an additional model with participant pool as a covariate and all possible interaction terms involving participant pool (see Supplementary Discussion).

In the Supplementary Information we provide more details regarding these analyses on page 2:

As participants were recruited via Leiden University and via Prolific, we examined, for each multilevel (logistic) model reported in the manuscript and in the Supplementary Information, whether participant pool mattered for our main results and conclusions. To this end, we computed additional models with participant pool as a covariate, and with interaction terms between participant pool and our experimental manipulations. In total and across dependent variables, this resulted in 59 possible interactions including participant pool. Because we had no a priori hypotheses, we applied Bonferroni-correction to avoid Type I errors (i.e., with $p = 0.05$ and 59 tests, interaction terms with $p < 0.0009$ are considered informative). Four interaction terms were significant, indicating differences in behaviour between our samples. First, LL types were paired more often to each other in the partner choice condition in the Prolific compared to the Leiden University pool ($z = 4.71$, $b_{LL \times platform} = 1.89$, $p < 0.0001$, 95% CI [1.26, 2.55]; $z = -3.43$, $b_{LL \times condition \times platform} = -1.90$, $p = 0.0006$, 95% CI [-2.63, -1.12]; base model in Supplementary Table 1). Second, cooperation in the partner choice condition increased over time in the 5 groups recruited via Prolific and decreased over time in the 16 groups recruited via Leiden University ($t(7724) = 7.04$, $b_{round \times platform} = 0.89$, $p < 0.0001$, 95% CI [0.64, 1.14]; base model in Supplementary Table 8). Likewise, cooperation of one's partner increased over time in the 5 groups recruited via Prolific, and decreased over time in the 16 groups recruited via Leiden University ($t(2898) = 5.59$, $b_{round \times platform} = 1.05$, $p < 0.0001$, 95% CI [0.68, 1.42]; base model in Supplementary Table 10).

In the Results section of our manuscript, we now report when behaviour differed between platforms. On page 7, lines 4-9:

As a result of these partner preferences, the most prevalent pairing in the partner choice condition consisted of two HH types being paired together; in 73% of the rounds an HH type was paired with another HH type (Fig. 2b). Although LL types were least preferred by all, the second most prevalent pairing was between two LL types; in 66.9% of the rounds an LL type was paired with another LL type (Fig. 2b; note that this pattern was more pronounced in the 16 groups recruited via Leiden University, than in the five groups recruited via Prolific, see Methods and Supplementary Information).

On page 8, lines 11-22:

Although partner choice segregated the population by type, it also produced higher overall cooperation compared to the assigned partner condition (Fig. 4a). Specifically, relative cooperation (the average contributions to the public good as a percentage of participant's individual endowment) decreased over time in the assigned partner condition (MLM, $t(7726) = -4.38$, $b_{condition \times round} = -0.33$, $p < 0.001$, 95% CI [-0.48, -0.18], Supplementary Table 8), but we found no credible evidence that cooperation changed in the partner choice condition (MLM, $t(7726) = -1.54$, $b_{round} = -0.08$, $p = 0.125$, 95% CI [-0.19, 0.02]; Supplementary Table 8). These results, however, should be interpreted with some caution, because exploratory analyses showed that in the five groups recruited via Prolific there is an increase in cooperation over time in the partner choice condition, whereas there is a decrease in the 16 groups recruited through Leiden University (see Supplementary Information). Nonetheless, findings resonate with previous work on partner choice showing that relative cooperation remained more stable under partner choice, possibly because participants could avoid uncooperative partner types.

Although we do agree it is valuable to understand whether behaviour is consistent across platforms, we also like to point out that we believe that any interaction effect need to be interpreted with caution, given the relatively small number of groups that participated via the Prolific platform. Instead, we believe that having collected data via different platforms is also beneficial as it increases the diversity of our sample and thus enhances the generalizability of our findings.

Third, a really nit-picking point, but the evidence for stake size in Dictator Games is not as mixed as the authors claim, given that two meta-analyses show a small effect of stakes, and that one of the two “no effect” papers the authors cite is already included in those meta-analyses. (The other is too new for those meta-analyses, and has its own problems for testing this question, like only paying some participants, which reduces the stakes and thus the effect size of stakes.) So this should be toned down rather than simply saying the results are mixed – they results clearly lean towards one side. Further, the authors should cite the Engel 2011 meta-regression (Experimental Economics), which combines many, many Dictator Game studies to show less generosity at higher stakes. This would represent a more accurate picture of the literature on stakes in Dictator Games.

Authors’ Response: Thank you for this feedback. We changed the relevant sentences in the Discussion, removed the 2 citations mentioned in the review, and included the citation of Engel (2011) (page 12, lines 19-23):

For instance, while stake size can impact generosity in Dictator Games^{30,31}, stake size did not affect decision-making in Ultimatum Bargaining³⁰, and studies on the effect of in-group membership²² and trust³² on cooperation showed that incentivized decisions did not differ from hypothetical ones. Accordingly, we expect current findings to generalize to situations with stronger incentives.